EMBO
Molecular Medicine

# *RARRES3* suppresses breast cancer lung metastasis by regulating adhesion and differentiation

Mònica Morales[1,†], Enrique J Arenas[1,†], Jelena Urosevic[1], Marc Guiu[1], Esther Fernández[1],
Evarist Planet[2], Robert Bryn Fenwick[3], Sonia Fernández-Ruiz[4], Xavier Salvatella[3,5], David Reverter[6],
Arkaitz Carracedo[4,7,8], Joan Massagué[9,10] & Roger R Gomis[1,5,*]

## Abstract

In estrogen receptor-negative breast cancer patients, metastatic relapse usually occurs in the lung and is responsible for the fatal outcome of the disease. Thus, a better understanding of the biology of metastasis is needed. In particular, biomarkers to identify patients that are at risk of lung metastasis could open the avenue for new therapeutic opportunities. Here we characterize the biological activity of *RARRES3*, a new metastasis suppressor gene whose reduced expression in the primary breast tumors identifies a subgroup of patients more likely to develop lung metastasis. We show that *RARRES3* downregulation engages metastasis-initiating capabilities by facilitating adhesion of the tumor cells to the lung parenchyma. In addition, impaired tumor cell differentiation due to the loss of RARRES3 phospholipase A1/A2 activity also contributes to lung metastasis. Our results establish *RARRES3* downregulation as a potential biomarker to identify patients at high risk of lung metastasis who might benefit from a differentiation treatment in the adjuvant programme.

**Keywords** breast cancer; lung metastasis; metastasis suppressor
**Subject Categories** Cancer; Stem Cells

## Introduction

Breast cancer (BC) is a highly heterogeneous disease, and there is clinical evidence of distinct patterns of disease relapse (Kennecke *et al*, 2010). In fact, the capacity of metastatic BC cells to grow in diverse environments may give rise to metastatic speciation, as indicated by the coexistence of tumor cells with distinct organ tropisms (bone, lung, liver, and brain) in patients with advanced BC (Nguyen *et al*, 2009). Analysis of gene expression profiles in experimental models of estrogen receptor-negative (ER⁻) BC contributed to identifying potential genes regulating or initiating lung metastasis (Minn *et al*, 2005; Eckhardt *et al*, 2012). Among the set of genes whose expression in breast tumor is associated with lung relapse, several encoded cytokines or secreted products that supported transendothelial migration from circulation into the lung parenchyma (Gupta *et al*, 2007; Padua *et al*, 2008). Additional genes, such as the extracellular matrix protein TNC, support the critical stem and progenitor cell pathways NOTCH and WNT and the viability of metastatic cancer cells in the lungs (Oskarsson *et al*, 2011). Interestingly, gene signatures associated with poor prognosis or site-specific metastasis indicate that relevant rearrangements in aggressive tumors and metastatic cells may also involve gene silencing (van't Veer *et al*, 2002; Minn *et al*, 2005; Lo *et al*, 2010; Cancer Genome Atlas Network, 2012). The silenced genes may encode several potential metastasis suppressors, responsible for the inhibition of overt metastasis at a secondary organ without affecting tumor growth at the primary site (Horak *et al*, 2008). *RARRES3*, a member of the lung metastasis gene signature (LMS) previously described (Minn *et al*, 2005), was identified in this group of genes as a potential metastasis suppressor.

The description of metastasis as an orderly sequence of basic steps—local invasion, intravasation, survival in circulation, extravasation, matrix remodeling, reinitiation, and colonization—has helped to rationalize the complex set of biological properties that must be acquired in order for a particular malignancy to progress toward overt metastatic disease (Vanharanta & Massague, 2013). In addition to acquiring motility properties, adapting adhesion

1  Oncology Program, Institute for Research in Biomedicine (IRB Barcelona), Barcelona, Spain
2  Biostatistics and Bioinformatics Unit, Institute for Research in Biomedicine (IRB Barcelona), Barcelona, Spain
3  Joint BSC-IRB Research Programme in Computational Biology, Institute for Research in Biomedicine (IRB Barcelona), Barcelona, Spain
4  CIC bioGUNE, Bizkaia Tecnology park, Derio, Spain
5  Institució Catalana de Recerca i Estudis Avançats (ICREA), Barcelona, Spain
6  Departament de Bioquímica i de Biologia Molecular, Institut de Biotecnologia i de Biomedicina, Universitat Autònoma de Barcelona, Bellaterra, Spain
7  Biochemistry and Molecular Biology Department, University of the Basque Country (UPV/EHU), Bilbao, Spain
8  Ikerbasque, Basque Foundation for Science, Bilbao, Spain
9  Cancer Biology and Genetics Program, Memorial Sloan-Kettering Cancer Center, New York, NY, USA
10 Howard Hughes Medical Institute, Chevy Chase, MD, USA
   *Corresponding author. Tel: +34 934039959; Fax: +34 934034848; E-mail: roger.gomis@irbbarcelona.org
   †The authors equally contributed to the work.

capacity, and remodeling the new microenvironment to enable metastasis, cancer cells also turn off differentiation programmes (Yang & Weinberg, 2008) and secure stemlike properties (Mani et al, 2008). In mouse models of BC and in patient samples, the loss of expression of differentiation markers correlates with tumor progression and metastasis (Kouros-Mehr et al, 2008a; Yu et al, 2012). GATA3 transcription factor determines luminal epithelial cell differentiation in the mammary gland (Kouros-Mehr et al, 2008b). Additionally, reduced GATA3 expression is strongly predictive of BC poor prognosis (Mehra et al, 2005) due to decreased cellular differentiation (Kouros-Mehr et al, 2008a) and increased tumor-initiating capacity (Asselin-Labat et al, 2011). However, it is unknown whether genes that promote the maintenance of differentiation attributes in basal-like ER$^-$ BC tumors restrain the malignant phenotype and metastatic dissemination by limiting metastasis-initiating capacity.

RARRES3 is a small protein with phospholipase $A_{1/2}$ ($PLA_{1/2}$) activity, responsible for producing signaling lipid secondary messengers in the form of arachidonic and eicosanoid derivatives (Han et al, 2010). Interestingly, RARRES3 was identified as a retinoic acid responder gene, and its expression was proposed to cause $G_0$ growth arrest in BC cells (DiSepio et al, 1998). Retinoic acid, a regulator of gene transcription and an inducer of cellular differentiation, has long been associated with differentiation patterns in both normal and cancer cells, with particular impact on certain hematopoietic malignancies (Grimwade et al, 2010). In this context, high expression of aldehyde dehydrogenase ALDH1A1, an enzyme that catalyzes the oxidization of retinol to retinoic acid (Marchitti et al, 2008), has been linked to retinoid metabolism and the attenuation of self-renewal capacity in normal hematopoietic stem cells (Chute et al, 2006). Similarly, it has been suggested that BC cells that retain tumor-initiating capacity select for the loss of expression of ALDH1A1 (Ginestier et al, 2007, 2009). Given the putative condition of RARRES3 as a responder gene to retinoic acid and its intrinsic catalytic activity (DiSepio et al, 1998; Han et al, 2010), the association of RARRES3 silencing in primary tumors with an increased lung metastatic activity is intriguing.

On the basis of these lines of evidence, we investigated whether cancer cells expressing RARRES3 have a selective disadvantage for metastasis, in particular in the lung microenvironment. Using BC cells, here we show that RARRES3 protein inhibits lung metastasis at two levels. First, RARRES3 blocks adhesion to the lung parenchyma and, second, the phospholipase activity of RARRES3 stimulates differentiation attributes, thus blunting metastasis-initiating functions at the lung required for the ER$^-$ BC cells to establish a lesion.

# Results

## RARRES suppression in breast tumors

RARRES3 is among the lung metastasis gene set whose mRNA expression level in breast tumors is associated with relapse to the lungs (Minn et al, 2005). In particular, in highly metastatic populations to the lung, RARRES3 mRNA is downregulated (Minn et al, 2005), thereby suggesting a potential metastasis suppressor function. To study this relationship, we confirmed the inverse association of RARRES3 expression with lung metastasis previously described in the MSKCC primary breast cancer set ($n = 82$) and, particularly, in those tumors defined as positive according to the lung metastasis signature (LMS) (Minn et al, 2005) (Fig 1A). Furthermore, our analysis was increased to cover a primary BC set including 560 patient samples with annotated clinical follow-up (MSK/EMC BC tumor dataset) (Bos et al, 2009) (details on the dataset in Supplementary Materials and Methods). The reduced expression of RARRES3 in primary tumors was significantly associated with the risk of lung metastasis (Fig 1B). Since low expression of RARRES3 strongly correlates with a higher propensity to develop lung metastasis (Fig 1B), and because RARRES3 levels vary widely between ER$^+$ versus ER$^-$ samples, we analyzed the effect of RARRES3 separately in the two tumor sets. This was particularly relevant given that ER status is a strong determinant of lung metastasis-free survival in BC patients (Supplementary Fig S1A). On the basis of ER status, we show that the inverse association of RARRES3 expression with high probability of lung metastatic disease is specific for the ER$^-$ tumor set (Fig 1C). Moreover, within the ER$^-$ subgroup, RARRES3 expression levels were exclusively inversely associated with risk of lung metastasis, but were not associated with the risk of bone or brain colonization (Supplementary Fig S1B and C). To date, compelling evidence associates high risk of BC relapse only with loss of expression of the metastasis suppressors PEBP1, NM23-H1, and IRF5. NM23-H1 has been proposed to act as a general metastasis suppressor in various tumor types (Marino et al, 2013), while PEBP1 and IRF5 have been described as bona fide metastasis suppressor genes in BC (McHenry et al, 2008; Li et al, 2009; Bi et al, 2011). Interestingly, PEBP1 expression levels are decreased in primary tumors (MSK/EMC dataset) that relapse to brain and lungs, thereby confirming the accuracy of our analysis, while RARRES3 levels in these clinical samples have prognostic value exclusively for the prediction of lung metastasis (Supplementary Table S1). In summary, these analyses highlighted RARRES3 as a putative key lung metastasis suppressor whose expression is reduced in primary BC tumors.

## RARRES3 prevents breast cancer lung metastasis

We studied the functional role of RARRES3 in experimental models of BC metastasis to lung. We used the metastatic BC cell line MDA-MB-231-LM2 (LM2), which was selected in vivo on the basis of a high capacity to colonize the lungs in mice, and the corresponding parental cell line MDA-MB-231, namely parental cells (Minn et al, 2005). LM2 cells showed a fivefold lower RARRES3 expression than their parental counterparts (Supplementary Fig S2A and B) and have been described to rapidly colonize the lungs when inoculated orthotopically in the mammary fat pad of immunodeficient mice (Padua et al, 2008). We examined how RARRES3 overexpression (Supplementary Fig S2A, B and C) modified the capacity of LM2 cells to colonize the lungs (Fig 2A, B and C). Of note, modulation of RARRES3 levels did not significantly alter the expression of any other LMS gene in parental or LM2 cell derivatives (Supplementary Fig S2A and B). In detail, Mock and RARRES3-overexpressing LM2 cells were injected into the mammary fat pad (MFP) of BALB/c Nude mice, and tumors were allowed to grow until they reach 300 mm$^3$. The tumors were then surgically resected, and lung colonization was allowed to develop (Fig 2A). Seven days after

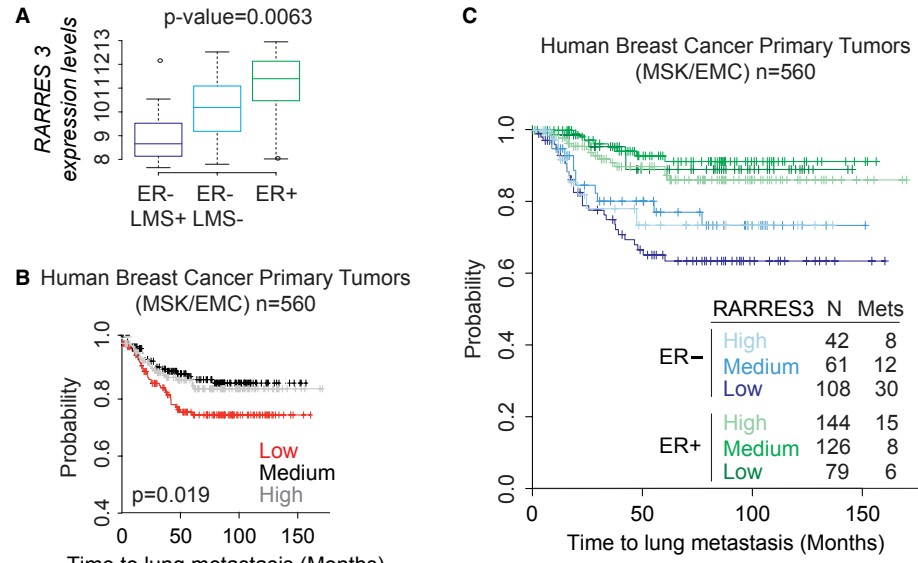

**Figure 1. RARRES3 suppression in breast tumors.**

A  Box plot of *RARRES3* expression levels in the MSKCC (*n* = 82) breast cancer tumor dataset according to ER and lung metastasis signature (LMS) status.

B  Kaplan–Meier representation of the probability of lung metastasis-free survival in the MSK/EMC breast cancer tumor dataset (*n* = 560) according to *RARRES3* levels of expression. Low, medium, and high represent *RARRES3* expression levels in the following way: low (< mean ± SD), medium (≥ mean ± SD and ≤ mean ± SD), and high (> mean ± SD).

C  Kaplan–Meier representation of the probability of lung metastasis-free survival in 560 breast cancer cases according to the ER status and *RARRES 3* expression levels according to (B).

mastectomy of the primary tumor, we assayed metastatic activity by bioluminescence imaging (BLI) of luciferase-transduced LM2 cells both in live animals and in the lungs *ex vivo* (Fig 2B). While six out of eight mice inoculated with LM2-Mock cells presented luciferase activity in the lungs, bioluminescence was detected in only two out of nine animals injected with LM2-RARRES3 cells (Fig 2B). Moreover, the amount of luciferase detected differed significantly, as shown in representative *in vivo* and *ex vivo* images of the lungs (Fig 2B). The resulting metastatic lesions showed positive staining for human Vimentin by immunohistochemistry (IHC), which specifically stains human MDA-MB-231 cells (Fig 2C). Several metastatic foci were observed throughout the lungs of mice bearing LM2-Mock tumors, while these were hardly observed in mice bearing LM2-RARRES3 tumors. Interestingly, *RARRES3* expression did not provide any growth advantage to cells when implanted at the MFP, as tested in an independent experiment (Fig 2D), or *in vitro* (Supplementary Fig S3A). RARRES3-expressing tumors did not display any change in vascular permeability, measured as effusion of intravenously injected rhodamine-conjugated dextran into the tumor or changes in VEGF expression levels (Supplementary Fig S3B and C). In addition, RARRES3 expression in primary tumors did not lead to differences in the number of circulating tumor cells, as measured by relative levels of human GAPDH to murine B2M (Supplementary Fig S3D). This observation suggests that the early steps of metastasis, including tumor vascularization and intravasation, were not under the influence of *RARRES3* expression.

Next, we focused on the late steps of metastasis with an emphasis on lung colonization. To this end, we examined the effect of RARRES3 restoration or depletion on lung metastatic colonization in LM2 or parental MDA-MB-231 cells, respectively (the latter, by

means of two independent short hairpin RNAs (Supplementary Fig S2A and B). We injected $2 \times 10^5$ cells into the lateral tail vein (TV) of BALB/c Nude mice and monitored lung colonization over time. Five days after cancer cell inoculation, the lung colonization signal was reduced in cells expressing high levels of *RARRES3*, compared to their counterparts, while growth upon this point was paralleled in all groups (Fig 3A and B). The overexpression of *RARRES3* greatly reduced the photon flux in the lungs of mice injected with LM2 cells (Fig 3A); this effect correlated with decreased lung colonization, as observed in H&E sections (Fig 3A). Concurrently, RARRES3-depleted parental cells exhibited enhanced capacity to colonize the lungs (Fig 3B). No differences in proliferation were observed among different groups, as measured by Ki-67 staining (Fig 3A and B), thereby suggesting that proliferation did not account for the differences observed at the metastatic site. Similarly, although apoptosis was diminished in lung lesions arising from LM2 cells when compared to parental ones, the modulation of *RARRES3* expression levels did not affect the amount of activated Caspase-3 in lung lesions or at the primary tumor site (Supplementary Fig S4A). Next, to generalize our findings, we validated the contribution of RARRES3 to lung colonization in patient-derived CN37 cells (Gomis *et al*, 2006) and in 4T1 mouse-spontaneous ER⁻ metastatic BC cells (Aslakson & Miller, 1992). CN37 and parental MDA-MB-231 cells show similar *RARRES3* expression, which we effectively downregulated (Supplementary Fig S4B). CN37 cells showed a low metastatic propensity to colonize the lungs. Control mice remained free of disease for 24 weeks (Fig 3C). However, RARRES3-depleted CN37 cells were able to initiate new lesions after a long latency period, and by week 20, lung colonization was observed in half the animals injected with CN37 shRARRES3 #1 and #2 cells (Fig 3C). Lesions

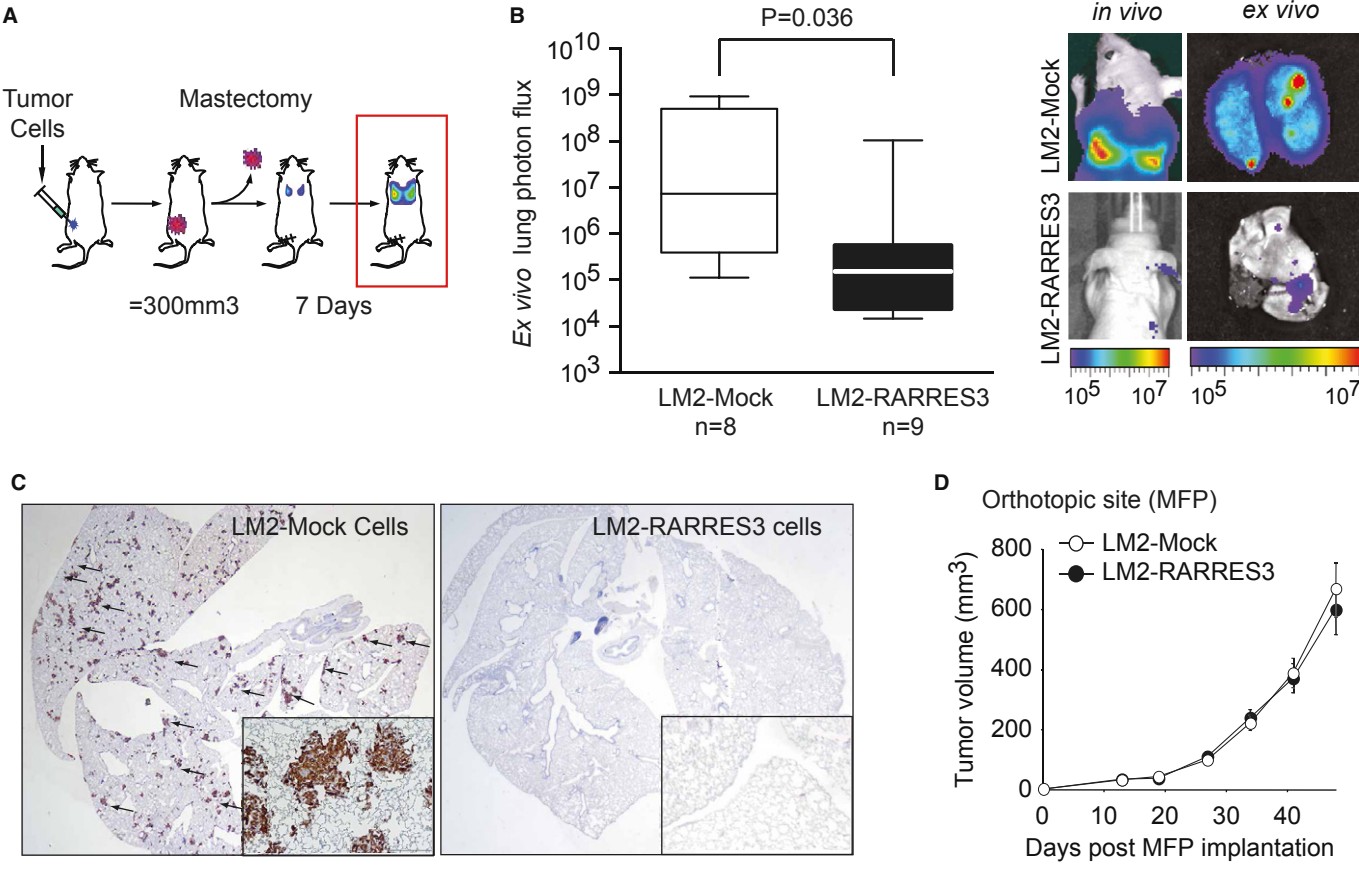

**Figure 2. RARRES3 prevents breast cancer lung metastasis.**

A   Schematic representation of *in vivo* experimental procedure to evaluate lung metastatic potential from the orthotopic site. The indicated cell lines ($5 \times 10^5$ cells) were injected contralaterally into the fourth mammary fat pad of mice. Tumors reaching 300 mm$^3$ were surgically removed. Seven days post-mastectomy, lung metastasis burden originated from size-matched tumors was quantified.

B   (Left panel) Quantification of *ex vivo* bioluminescent signal at the lungs in each experimental group subjected to the tumor growth/resection scheme described in (A) at end point. $n = 8$ and $n = 9$ mice per group were used. Whiskers plots from min–max values were used. (Right panel) Representative bioluminescence images of *in vivo* and *ex vivo* lung colonization of the mice are shown.

C   Representative human Vimentin IHC staining of whole lung sections to highlight metastatic tumor lesions from (B) are shown. Inset panels (4× magnification) reflect the size and multiple metastatic foci detected in the LM2-Mock group.

D   LM2 cells ($5 \times 10^5$) expressing an empty vector (LM2-Mock) or a RARRES3-expressing vector (LM2-RARRES3) were injected contralaterally into the fourth mammary fat pad of mice, and tumor growth was measured over time. $n = 20$ per group. Data are averages $\pm$ SEM.

caused by *RARRES3*-depleted CN37 cells continued to grow until week 24 (Supplementary Fig S4C). As observed in parental MDA-MB-231 cells, RARRES3 levels did not affect the capacity of CN37 cells to grow at the primary site, determined 10 weeks post-inoculation (Supplementary Fig S4D). In contrast, 4T1 cells show reduced *RARRES3* expression compared to parental MDA-MB-231 cells, which we increased by means of exogenous expression (Supplementary Fig S4E). 4T1 cells showed a high metastatic propensity to colonize the lungs in syngeneic BALB/c mice, which developed overt lung metastasis 20 days after inoculation. RARRES3-expressing 4T1 cells displayed a significant reduction in the capacity to colonize the lung 20 days post-injection (Fig 3D).

Interestingly, human Vimentin IHC revealed significant increase in lung metastatic foci when cells with low *RARRES3* expression were inoculated (LM2 cells or RARRES3-depleted parental cells) compared to populations expressing high levels of *RARRES3* (parental cells and RARRES3-expressing LM2) (Fig 3E). Although the area

of the lesions, as expected, was larger for LM2 cells than parental ones, no differences were observed in LM2 or parental cells expressing different *RARRES3* levels in comparison with their respective controls (Fig 3A and B). This observation was consistent with the lack of differences in proliferation or apoptosis, as measured by Ki67 or caspase-3 activity, caused by variations in *RARRES3* expression (Fig 3A and B and Supplementary Fig S4A). The above results suggest that *RARRES3* expression prevents lung colonization initial steps.

## RARRES3 suppresses metastatic cell adhesion to the lung parenchyma

Next, we addressed the mechanism by which RARRES3 may prevent lung metastasis. We initially investigated whether RARRES3 regulates apoptosis in the circulation or at the metastatic site. Under the former scenario, cell death may occur by anoikis due to the absence

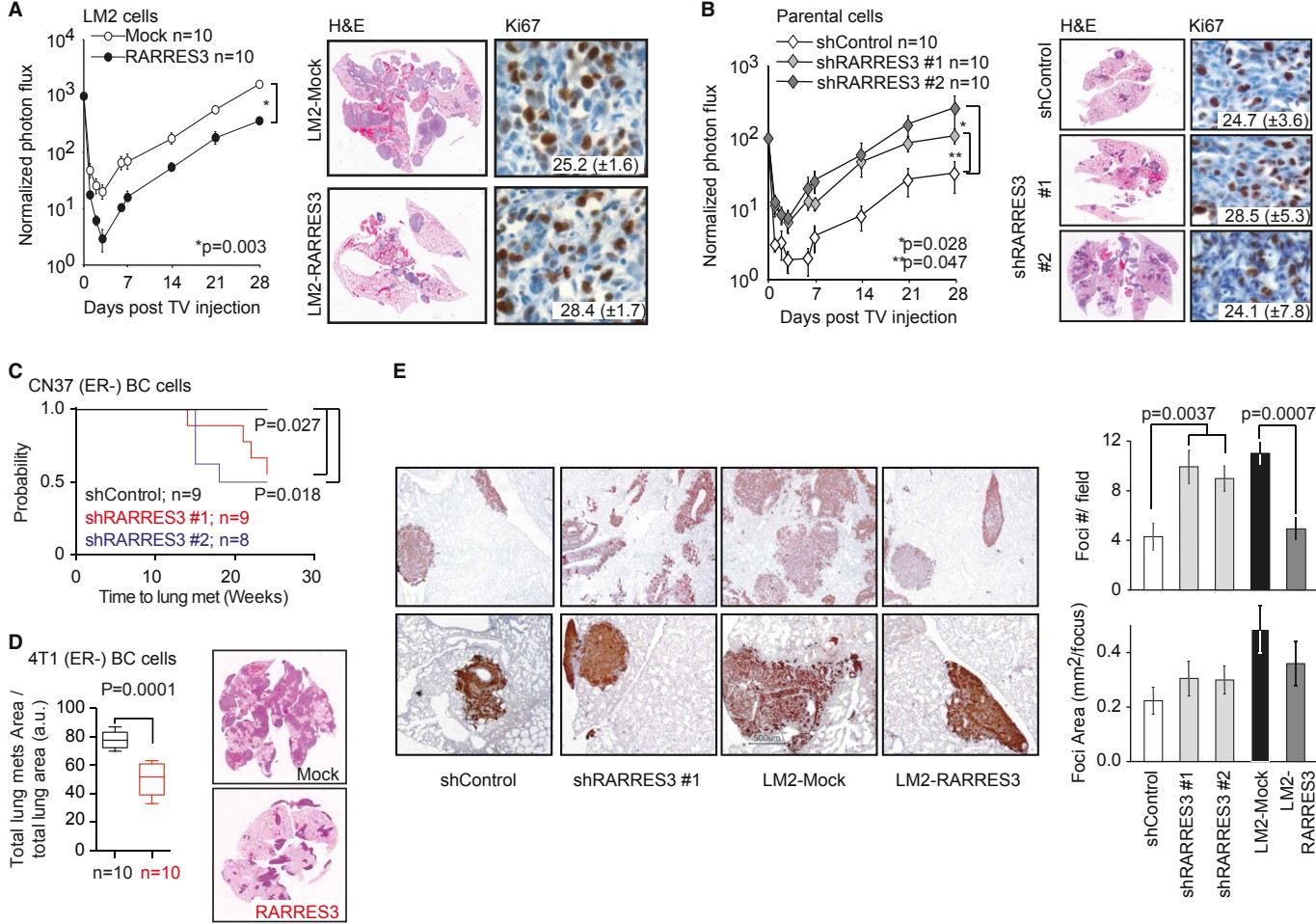

**Figure 3. RARRES3 depletion facilitates lung colonization.**

A   LM2-Mock and LM2-RARRES3 cells ($2 \times 10^5$) were injected into the tail vein of mice. Lung colonization was assayed by weekly bioluminescence imaging. Plots show normalized photon flux in the lung over time ($n = 10$ per group). Representative H&E and Ki-67 staining of lung sections 4 weeks after engrafting are shown. Data are averages $\pm$ SEM.

B   Parental MDA-MB-231 cells ($2 \times 10^5$) transduced with a control vector (shControl) or two independent *RARRES3* shRNA vectors (shRARRES3 #1 and #2) were injected into the tail vein of mice. Lung colonization was assayed by weekly bioluminescence imaging. Plots show normalized photon flux in the lung over time ($n = 10$ per group). Representative H&E and Ki-67 staining of lung sections 4 weeks after xenografting are shown. Data are averages $\pm$ SEM.

C   CN37 patient-derived metastatic breast cancer cells ($2 \times 10^5$) transduced with a control vector (shControl) or two independent *RARRES3* shRNA vectors (shRARRES3 #1 and #2) were injected into the tail vein of mice ($n = 8$, 9 and 9 per group, respectively). Lung colonization was assayed by weekly bioluminescence imaging. Kaplan–Meier curve of the probability of lung metastasis-free survival for CN37 shControl, shRARRES3#1, and #2 is presented. Log-rank test was used to establish statistical significance.

D   4T1 mouse-derived breast cancer cells ($2 \times 10^5$) transduced with a Mock or *RARRES3* vector were injected into the tail vein of mice ($n = 10$ per group). Lung colonization was assayed by calculating the total area of lung metastasis lesion normalized per the total area of the lungs (H&E). Three sections were analyzed per lung. Data are averages of 10 lungs (mice) per group $\pm$ SEM. Wilcoxon test was used to establish statistical significance. Representative H&E sections are shown.

E   Vimentin IHC of lung sections from animals inoculated with the indicated lines. Upper panels: 2× magnification. Lower panels: 4× magnification. Quantification of the number of foci per field and of the average area per foci is shown in the right panels. Data are averages $\pm$ SEM.

of cell attachment (Nagaprashantha *et al*, 2011). Anoikis, tested *in vitro* by culturing cells in suspension, was reduced in LM2 cells compared to the parental MDA-MB-231 line; however, RARRES3 downregulation or overexpression did not alter the fraction of cells that succumbed to the lack of cell attachment (Fig 4A). Similarly, we tested apoptosis *in vivo* 6 h post-injection, when cells were trapped at the lung vasculature but had not yet extravasated. Apoptosis was assessed by injection of a luciferase ZVAD-protected prosubstrate susceptible to be activated only upon Caspase-3/7 activation in apoptotic cells. In concordance with the results obtained

*in vitro*, LM2 cells exhibited lower levels of apoptosis than parental cells, but RARRES3 did not modify the intensity of apoptosis in parental or highly metastatic populations (Fig 4A).

In the absence of a direct cellular pro-apoptotic effect, we hypothesized that RARRES3 instead controls the metastatic lung colonization steps of extravasation/adhesion and/or metastatic lesion initiation. First, we tested the contribution of RARRES3 to lung extravasation and adhesion *in vivo* using LM2 cells with and without RARRES3 overexpression. These cellular populations were injected into mice and 2 days later the number of cells extravasated

**A**

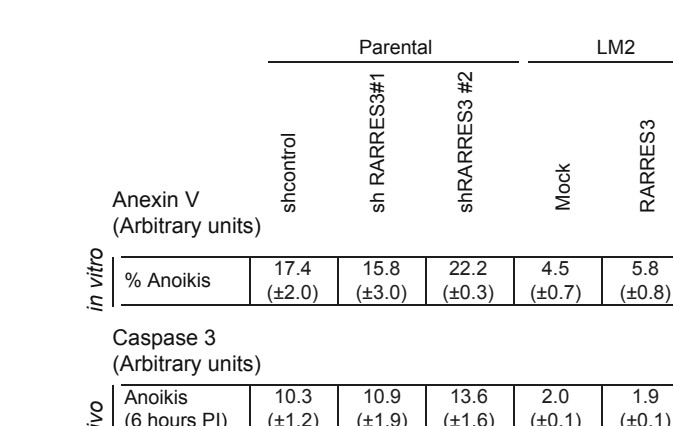

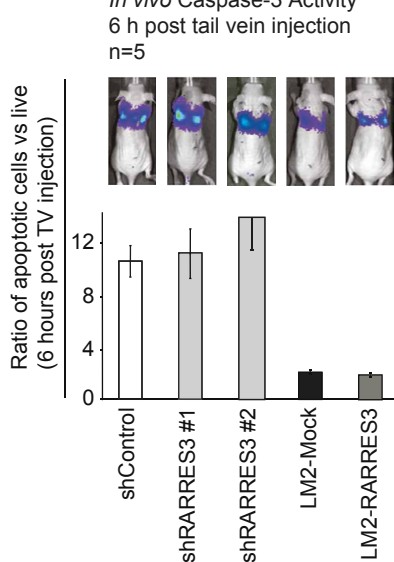

| | | Parental | | | LM2 | |
|---|---|---|---|---|---|---|
| | | shcontrol | sh RARRES3#1 | shRARRES3 #2 | Mock | RARRES3 |
| Anexin V (Arbitrary units) | | | | | | |
| *in vitro* | % Anoikis | 17.4 (±2.0) | 15.8 (±3.0) | 22.2 (±0.3) | 4.5 (±0.7) | 5.8 (±0.8) |
| Caspase 3 (Arbitrary units) | | | | | | |
| *in vivo* | Anoikis (6 hours PI) | 10.3 (±1.2) | 10.9 (±1.9) | 13.6 (±1.6) | 2.0 (±0.1) | 1.9 (±0.1) |
| | Lung Mets (7 weeks PI) | 17.4 (±5.4) | 21.6 (±2.2) | 27.1 (±4.7) | 8.9 (±2.0) | 9.3 (±1.9) |

*In vivo* Caspase-3 Activity
6 h post tail vein injection
n=5

**B**    *In vivo* Pulmonar Extravasation
48 h post tail vein injection

**C**    *In vitro* Adhesion

**D**    *In vitro* Adhesion

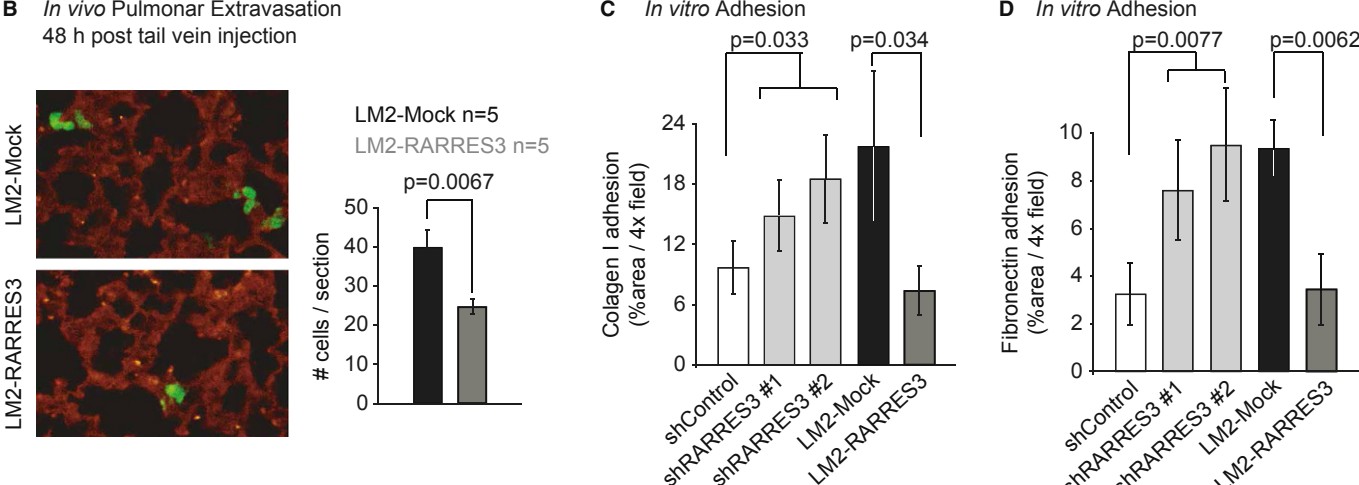

**Figure 4.  RARRES3 impairs metastatic cell adhesion to the lung parenchyma.**

A  (Left panel) Percentage of apoptotic cells under anoikis conditions, as determined by FACS measurement of Anexin V staining. Same number of cells of the indicated cell lines were plated, and measurements were taken at the indicated time point. Data are averages of three independent experiments ± SD. (Right panel) Apoptotic human luciferase activity of the indicated cell lines was measured *in vivo* using a Caspase-3 luciferin pro-substrate and normalized to total luciferase activity at the indicated time points after injection (*n* = 5 per group). Data are averages ± SEM. PI stands for post-injection.

B  Cell tracker green-labeled LM2-mock and LM2-RARRES3 cell lines were injected via the tail vein and allowed to lodge in the lungs. Two day post-injection, mice were inoculated with rhodamine-lectin and 30 min later were perfused with 5 ml of PBS to remove cells attached to the vasculature. Lungs were extracted, flushed with PBS, and fixed-frozen in OCT, and frozen sections were obtained. Representative confocal images of extravasated cells (green) to the lung parenchyma are shown. In red, vasculature staining using rhodamine-lectin. Right panel: Quantification of extravasated cells in each condition is plotted. *n* = 5 mice per group. 10 sections per mouse were scored. Data are averages ± SD.

C, D  Adhesion to collagen I and fibronectin, respectively. The MBA-MB-231 shControl, shRARRES3 #1 and #2, LM2-Mock and LM2-RARRES3 cells were labeled with cell tracker green and plated ($5 \times 10^4$) in triplicate in 24-well inserts coated with Collagen I (C) and Fibronectin (D). One hour post-plating, inserts were washed twice with PBS to remove non-attached cells and fixed in PFA. Images were taken, and the area covered by cells was determined using Image J. The percentage of area covered by cells in Collagen I (C) or Fibronectin (D) inserts is shown. Data are averages of three independent experiments ± SD (*n* = 3).

into the lung parenchyma was determined. While 39.75 (± 4.5) LM2-Mock cells were observed per lung section, the overexpression of *RARRES3* significantly reduced these levels to 24.5 (± 1.9) cells (Fig 4B), suggesting that alterations in features required for extravasation, such as migration through an endothelial barrier, invasion or

adhesion, could account for this observation. It has been previously reported that the capacity of LM2 cells to migrate through an endothelial cell layer is 5-fold greater than that of MDA-MB-231 parental cells (Gupta *et al*, 2007). However, RARRES3 expression levels did not affect the migratory capacity of either of these cell populations

(Supplementary Fig S5A). Similarly, invasion properties, measured as the capacity of cells to degrade and invade through a matrigel layer, were not affected by *RARRES3* expression levels, either in the presence of growth factors or in growth factor-reduced matrigel (Supplementary Fig S5B). In contrast, when we determined the affinity of cells for lung extracellular matrix proteins by measuring the adhesion to Type I collagen and fibronectin, significant differences were detected (Fig 4C and D). LM2-Mock cells in contact with a Type I collagen or fibronectin matrix exhibited enhanced adhesion compared to parental MDA-MB-231 shControl cells (Fig 4C and D). Downregulation of RARRES3 in MDA-MB-231 parental cells caused a marked increase in the capacity of cells to adhere to these two matrices (Fig 4C and D). Correspondingly, RARRES3 overexpression in highly metastatic LM2 cells reduced adhesion to Type I collagen and fibronectin to the levels shown by parental shControl cells (Fig 4C and D), confirming that RARRES3 expression attenuates cell adhesion to the lung parenchyma.

## RARRES3-PLA$_{1/2}$ catalytic activity stimulates differentiation

The emergence of metastasis reflects the capacity of cancer cells not only to overcome the need to adhere to the vasculature and extracellular matrix but also to initiate a new lesion. Our observation that high levels of *RARRES3* expression reduced the number of metastatic foci suggested that RARRES3 inhibits metastatic colony initiation. We hypothesized that RARRES3 blocks the initiation of metastatic lesions by promoting cellular differentiation signals through its intrinsic phospholipase A$_{1/2}$ catalytic activity. PLA$_{1/2}$ activity is pivotal for the production of the arachidonic and lyso-phospholipid precursors that result from the hydrolysis of the acyl chain of phospholipids (Wang & Dubois, 2010). Upon downstream modifications by cyclooxygenases, these precursors are modified to active compounds called eicosanoids (prostaglandins and leukotrienes), which may signal as lipid secondary messengers and promote differentiation (Wang & Dubois, 2010). We modeled the three-dimensional structure of RARRES3 based on the structure of the HREV107 family member and identified the key residues that form the PLA$_{1/2}$ catalytic domain of the human gene (Fig 5A and B), including His23, His35, Arg18, and Cys113. These four residues, conserved across species, comprise a well-defined catalytic core of the above-described enzymatic activity (Fig 5B and Supplementary Fig S6) (Uyama *et al*, 2009). By mutating two of the catalytic core amino acids (H23P and C113S), we confirmed that RARRES3 PLA$_{1/2}$ activity and its catalytic core residues were responsible for changes in cellular arachidonic acid content (Fig 5C).

Next, we evaluated the contribution of RARRES3 PLA$_{1/2}$ catalytic activity to cell differentiation processes through lipid signaling mediators such as arachidonic derivatives. Interestingly, Peroxisome proliferator-activated receptors (PPARs) are a group of nuclear receptor proteins that function as transcription factors and whose activity is dependent on arachidonic derivatives (Sertznig *et al*, 2007). PPARs play an essential role controlling cellular metabolism, development, and differentiation (Sertznig *et al*, 2007). We investigated whether PPAR function was sensitive to RARRES3 catalytic activity, by using a PPAR-specific luciferase reporter assay based on three copies of the rat acyl-CoA oxidase peroxisome proliferator response element, Aox-3x-PPRE-Luc. We found that RARRES3 expression increased the reporter transcription and a RARRES3-DEAD mutant

abrogated this effect (Fig 5D). To further evaluate the clinical relevance of the catalytic activity of RARRES3 and its association with differentiation markers in BC, we initially focused on 13 well-known PPAR target genes associated with differentiation processes (Sertznig *et al*, 2007), including lipid metabolism enzymes, fatty acid transport and uptake genes, the peroxisome maintenance gene, and gene transcription. We found that the expression of these genes significantly correlated with RARRES3 expression in ER$^-$ BC primary tumors (Fig 5E). Interestingly, the expression levels of six of these genes (PEX11A, ACOX2, ACAD8, HMGCS2, SLC27A2, and FABP5) were individually and significantly associated with risk of lung metastasis recurrence in these tumors (Fig 5E). Next, we performed a cross-validation of our group of PPAR-dependent RARRES3-correlated genes in the combined expression dataset of 211 clinically annotated human primary ER$^-$ breast tumors (MSK/EMC dataset). The outcome of interest was time to lung recurrence (TTR) after primary tumor surgical removal. Using gene set enrichment analysis (GSEA) (Subramanian *et al*, 2005), we found a strong negative association between the PPAR-dependent RARRES3-correlated gene set and an increased risk of lung recurrence upon therapeutic treatment (normalized enrichment score of −1.88 and a false discovery rate of 0.001) (Fig 5F). On the basis of these lines of evidence, we interrogated whether RARRES3 expression in primary ER$^-$ tumors also correlated with well-established mammary epithelial differentiation genes. With this aim and to determine the genes significantly associated with changes in *RARRES3* expression, we performed a correlation analysis between *RARRES3* and all the other genes in the MSK/EMC primary tumor expression dataset (Affymetrix U133Aplus2). RARRES3 expression correlated positively with the differentiation genes GATA2 and GATA3, and inversely with the EZH2 polycomb protein, a pluripotency marker gene (Fig 5G). Collectively, these observations strongly support the notion that the retention of RARRES3 expression and the production of signaling mediators and precursors through its PLA$_{1/2}$ activity are associated with BC tumors preserving some of their differentiation attributes.

## RARRES3 prevents metastasis colony initiation

A potential consequence of RARRES3 expression in BC cells is the retention of certain differentiation properties, which could challenge metastasis-initiating functions. To test this hypothesis, we measured the lung metastasis-initiating capacity of limiting dilutions of Mock and RARRES3-expressing LM2 cell populations upon intrapulmonary injections into BALB/c Nude mice. The cells were injected directly into the lung parenchyma (absence of extravasation/adhesion) as opposed to subcutaneously or in the MFP, since the extracellular matrix component of the lung metastatic niche has been reported to be crucial for lung metastasis-initiating capacity (Oskarsson *et al*, 2011). Tumor emergence was used as a surrogate of the metastasis-initiating capacity of cells. Mock and RARRES3-overexpressing LM2 cells inoculated in high numbers (5,000 cells) colonized the lungs with a similar frequency and latency (Supplementary Fig S7A). At lower dosages (500 or 50 cells), Mock LM2 cells retained the capacity to colonize the lungs with high efficiency, whereas RARRES3-expressing ones displayed a reduced capacity (Fig 6A). Similarly, at low dosages (500 and 50 cells), 4T1 murine metastatic cells expressing RARRES3 showed reduced metastatic initiation capacity in the lungs (Fig 6B and Supplementary Fig

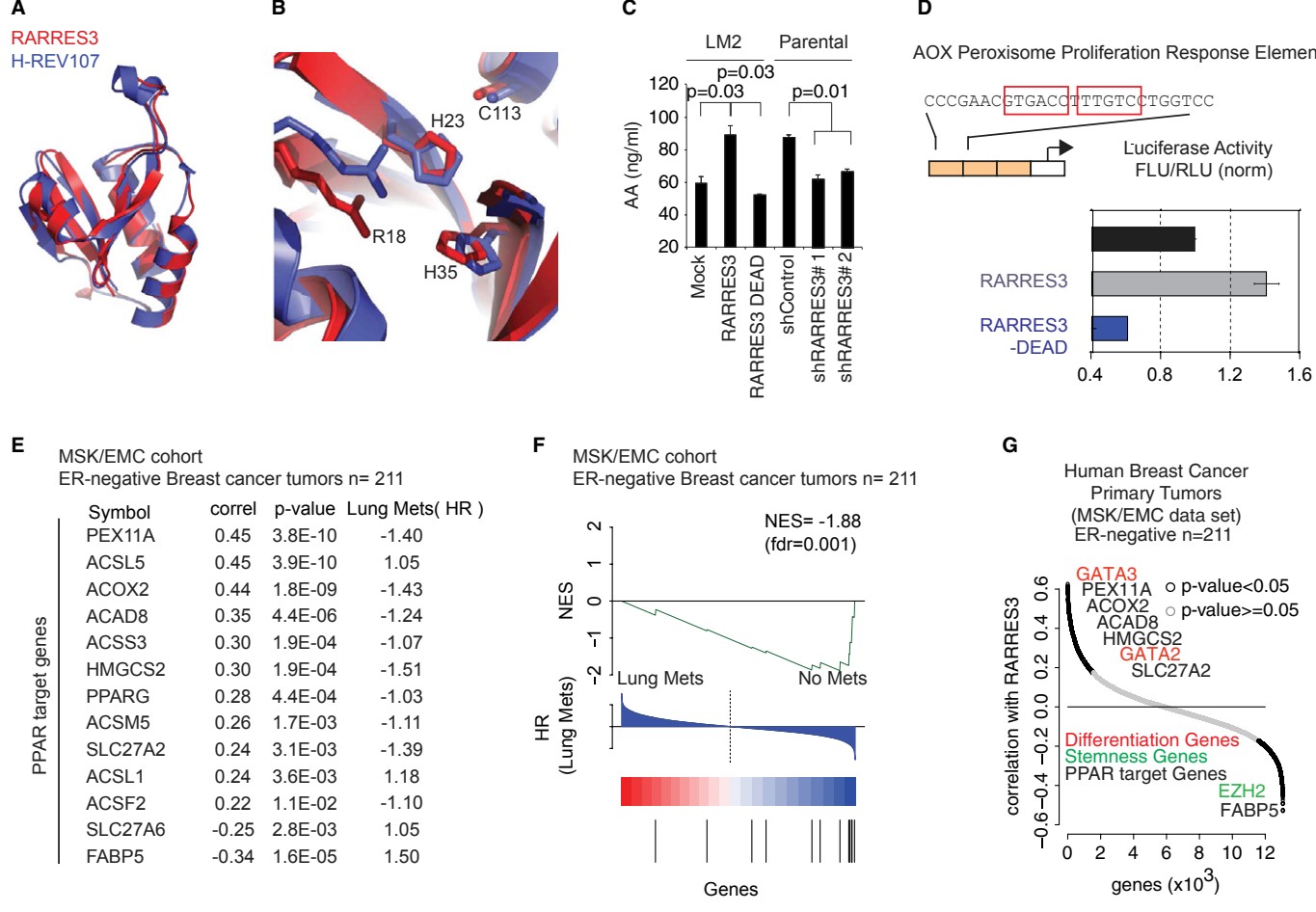

**Figure 5.  RARRES3 PLA$_{1/2}$ catalytic activity triggers differentiation-signaling mediators.**

A   Homology 3D structural model of RARRES3 (red) aligned with the structure of H-REV107 (violet) (Ren *et al*, 2010).

B   The conserved catalytic residues of RARRES3 and H-REV107 showing the proposed similarity of function.

C   Human Arachidonic Acid (AA) levels (ng/ml) measured in cell extracts of MDA-MB-231 Parental and LM2 cellular derivatives expressing different levels of RARRES3. Data are represented as the mean of three independent experiments ± SEM.

D   Luciferase activity of 3x-AOX peroxisome proliferation response element reporter plasmid in MDA-MB-231 parental cells transiently transfected with Mock, RARRES3- and RARRES3-DEAD-expressing vectors. Activity of 3xAOX promoter was normalized to control condition and presented in arbitrary units. Data are mean of three independent experiments with ± SD.

E   The correlation coefficient and significance of *RARRES3* expression levels in ER$^-$ BC primary tumors from the MSK/EMC meta-cohort against the expression of 13 PPAR target genes represented in the U133A affymetrix array are shown, together with the p value associated with each correlation. Moreover, the risk of lung metastasis (HR) associated with the expression of each of those genes in primary tumors is also reported.

F   Gene set enrichment analysis (GSEA) representing association of HR of lung metastasis with the PPAR target *RARRES3*-correlated gene set in the human breast cancer dataset (MSK/EMC expression dataset). NES-normalized enrichment score; FDR-false discovery rate; HR-hazard ratio.

G   The correlation coefficient of RARRES3 expression levels in ER$^-$ BC primary tumors from the MSK/EMC meta-cohort against the expression of all the genes represented in the U133A affymetrix array is shown. In red, differentiation GATA transcription factors. In black, some PPAR target genes described in (E). In green, stemness gene.

S7A). In contrast, increased frequency of lung colonization for RARRES3-depleted CN37 cells in low dosages (1,000 cells) was detected (Fig 6A). Therefore, the reduction in metastasis-initiating capacity paralleled the expression of RARRES3.

Next, we analyzed the contribution of *RARRES3* expression to the rate of oncosphere formation in 3D and 2D culture conditions, a readout of pluripotency (Dontu *et al*, 2003; Liao *et al*, 2007; Grimshaw *et al*, 2008) that determines the ability of a single BC cell to start a new colony. For this purpose, we grew the various RARRES3-expressing cell populations previously established

(MDA-MB-231, CN37, and 4T1 cells) in matrigel. 3D culture systems recapitulate organotypic growth with respect to a polarized phenotype, specialized cell–cell contacts, and attachment to an underlying basement membrane (Schmeichel & Bissell, 2003; Debnath & Brugge, 2005). All of these features are required for the proper control of cellular proliferation, survival, and differentiation. We seeded the various RARRES3-expressing LM2, CN37, and 4T1 cells into a 3D matrix as described in the experimental section. The number of colonies formed was quantified and compared in each cellular population (Fig 6C and Supplementary Fig S7B).

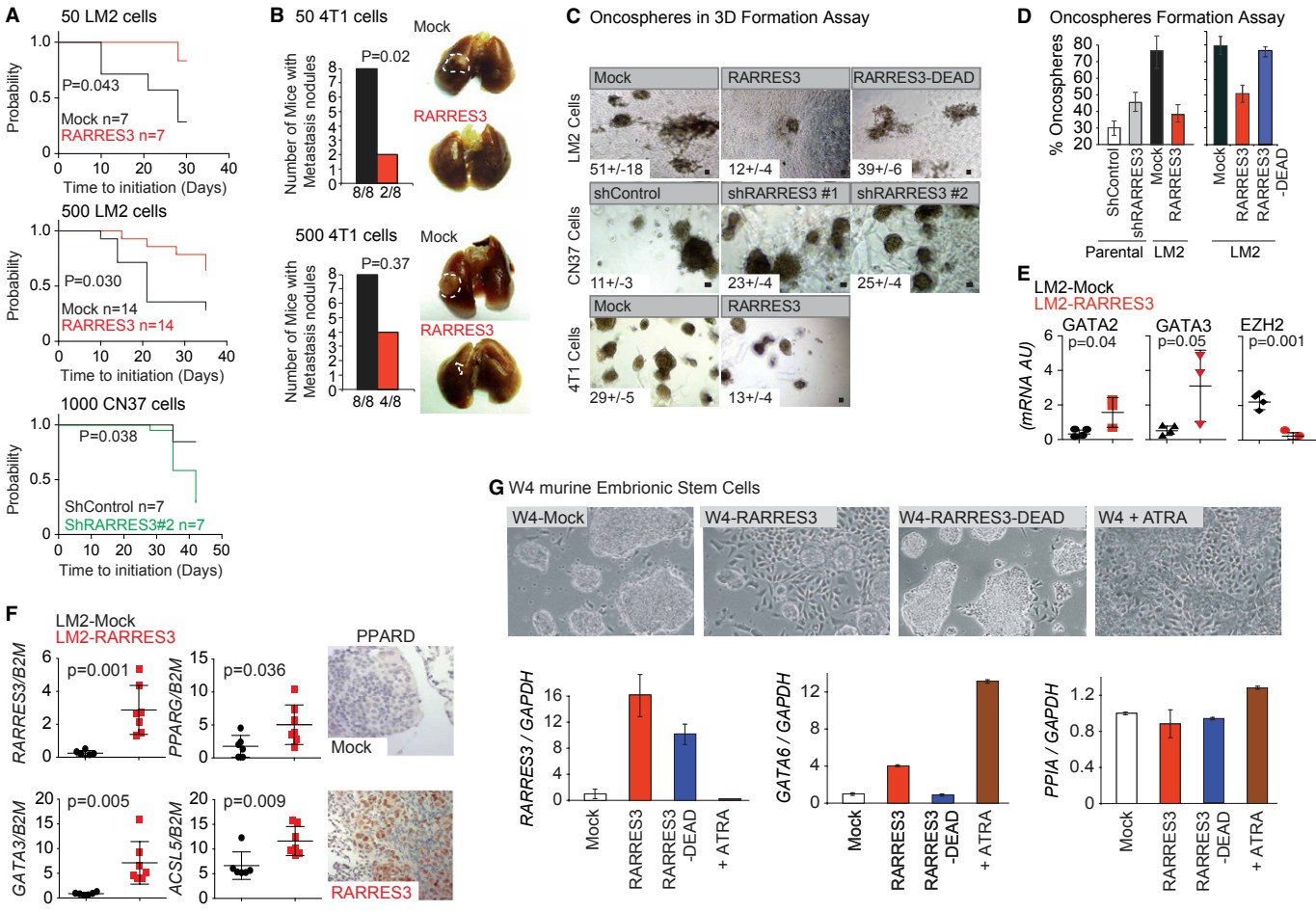

**Figure 6. RARRES3 induces cellular differentiation thus impairing metastasis initiation.**

A   Mock- and RARRES3-expressing LM2 cells or shControl and shRARRES3 CN37 cells at the indicated dilutions were intrapulmonary injected, and grow in the lungs assessed by luciferase bioluminescence over time. Kaplan–Meier plots of the probability of lung metastasis-free survival and log-rank test were used. *n* = 7 mice per group was used unless indicated otherwise.

B   Mock- and RARRES3-expressing 4T1 cells at the indicated dilutions were intrapulmonary injected, and grow in the lungs was confirmed macroscopically 20 days post-implantation. Contingency plots and Fisher exact test were used. Representative images are shown (white dashed line limits tumor area). *n* = 8 mice per group were used.

C   Equal limiting dilutions of the indicated cell lines were plated in Matrigel. The number of organotypic spheres per plate was counted. Shown is the average of three independent experiments ± SD. Representative images are shown. Scale bar represents 50 μm.

D   Limiting dilutions of the indicated cell lines were performed, and one cell was plated per well of 96-well plates. One plate per cell line was cultured. The percentage of wells that generated oncospheres was calculated. Shown is the average of three independent experiments ± SD.

E   Human mRNA qPCR analysis of the indicated genes and cell lines in oncospheres cultures is shown. Shown is the average of at least three independent experiments ± SD.

F   Human mRNA qPCR or IHC analysis of the indicated genes on lung metastasis of mice inoculated with Mock and RARRES3-expressing LM2 cells is shown (*n* = 7). Shown is the average all samples ± SD or a representative image of these tumors.

G   RW.4 cells expressing a control vector (W4), a RARRES3, or a RARRES3-DEAD mutant-overexpressing vector (W4-RARRES and W4-RARRES3-DEAD) were cultured in the presence or absence of 1M all-trans retinoic acid (ATRA) for 3 days. Upper panels: representative images of the cultures are shown. Lower panels: qPCR analysis of *RARRES3*, *GATA6*, and *PPIA* (control) mRNAs of the indicated cultures was performed. Data are average of three independent experiments ± SD.

Interestingly, increased *RARRES3* expression reduced the capacity of all the cell types to undergo organotypic growth. Catalytic RARRES3-DEAD mutant overexpression in LM2 cells did not abrogate the capacity of these cells to form 3D spheroids (Fig 6C and Supplementary Fig S7A). This observation suggests that the enzymatic activity of the protein is necessary for RARRES3 to prevent the initiation of the 3D structures from a single cell. Similarly, oncosphere formation from single cells in low-attachment plates was also tested in various RARRES3-expressing MDA-MB-231 cell

populations. One cell per well was seeded in a 96-well plate, and the ability to form oncospheres was determined 2 weeks later. While only 30.2% of parental shControl cells formed oncospheres, up to 76% of LM2 cells showed this potential (Fig 6D). *RARRES3* downregulation enhanced the capacity of the parental cell line to form oncospheres, while overexpression of RARRES3 dramatically abrogated this property in LM2 cells in a $PLA_{1/2}$ activity-dependent manner (Fig 6D). Moreover, the levels of *RARRES3* mRNA were decreased in oncospheres produced by MDA-MB-231 cells compared

to the respective original cells in culture (Supplementary Fig S7C). The reduction in the capacity of RARRES3-expressing LM2 cells to form spheroids was correlated with higher expression of the differentiation markers GATA3 and GATA2, associated with RARRES3 expression in primary tumors (Fig 6E). Similarly, lung metastatic tumors in mice originated from RARRES3-expressing LM2 cells also showed increased expression of some differentiation attributes, including the differentiation transcription factor GATA3, as well as some PPAR targets (PPARD, PPARG, and ACSL5) significantly correlated with RARRES3 expression in primary tumors (Fig 6F).

The maintenance of a stem phenotype is frequently associated with the expression of pluripotency genes and the absence of differentiation markers (Dontu *et al*, 2003; Sparmann & van Lohuizen, 2006; Chou *et al*, 2010). Our previous observations reinforced the notion that RARRES3 promotes differentiation. To test this hypothesis and assess the capacity of RARRES3 to engage differentiation, we used a pluripotent embryonic mouse cell line, RW-4, where differentiation can be easily monitored. Treatment of RW-4 cells with all-trans retinoic acid (ATRA) induces differentiation, a process that is controlled by GATA6, since its absence precludes differentiation (Capo-Chichi *et al*, 2005). Thus, GATA6 controls and also can be used as a marker of the differentiation status in these cells (Capo-Chichi *et al*, 2005). RW-4 cells were cultured in gelatin plates in the absence of feeders. Under these conditions, the cells grew in tight groups, and only a few isolated cells that accomplished differentiation presented a long shape and attached to the plate (Fig 6G). Overexpression of RARRES3 induced the differentiation of RW-4 cells, a phenotype that is easily detected by the presence of numerous differentiated cells attached to the plate and the size reduction of the groups of pluripotent cells (Fig 6G). This phenotype was abrogated in the absence of RARRES3 catalytic activity (Fig 6G). As a control,

we treated RW-4 cells with ATRA. As expected, almost all the cells in the plate engaged in differentiation (Fig 6G). To quantify the extent of differentiation induced by RARRES3, we analyzed the mRNA levels of GATA6 by quantitative PCR. RARRES3 induced the expression of GATA6 by up to fourfold, whereas ATRA induced an increase of 13-fold (Fig 6G). These results indicate that the catalytic activity of RARRES3 induces the differentiation of pluripotent mouse embryonic cells and is associated with the retention of differentiation markers in experimental systems of BC and primary tumors. These observations support the concept that RARRES3 prevents the initiation of lung metastatic lesions by enforcing the retention of differentiation features.

## Discussion

Here we provide novel evidence on the role of RARRES3 in preventing BC lung metastasis by the combined inhibition of metastatic adhesion and initiation. We have shown that RARRES3 impedes the adhesion of BC cells to the lung parenchyma while enforcing the retention of differentiation properties, thus restraining the adhesion and initiation of new lesions by the metastatic cells in the lungs (Fig 7).

RARRES3 expression did not cause any differences in primary tumor growth, angiogenesis, or proliferation. On the contrary, this metastasis suppressor modulated mainly steps required at the metastatic site, including metastatic initiation. The acquisition of low expression levels of *RARRES3* in ER⁻ BC primary tumors that metastasize to the lung is directly associated with a reduction in *GATA* differentiation genes (Chou *et al*, 2010) and inversely correlated with expression of the *EZH2* pluripotency gene marker (Sparmann &

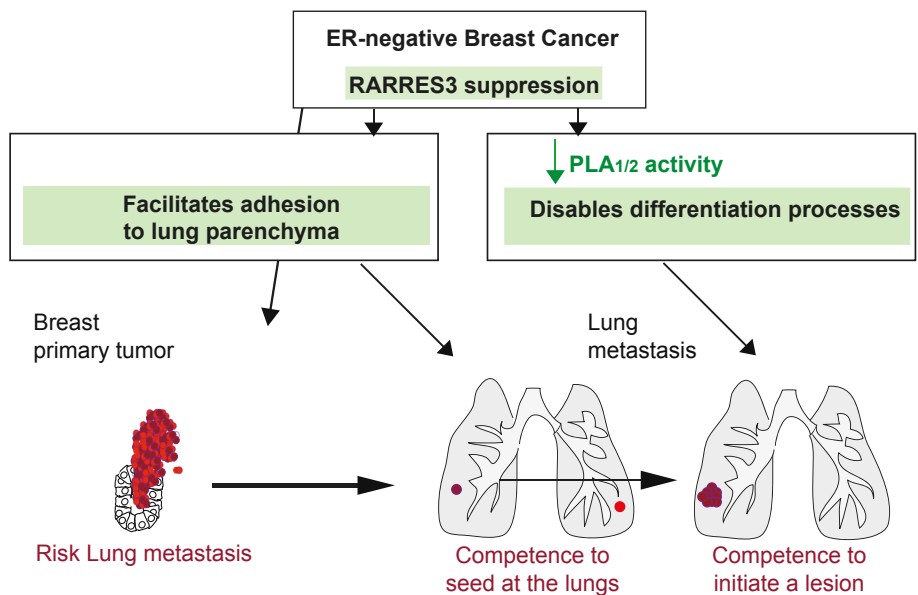

**Figure 7. Schematic model.**
Model showing how RARRES3 suppression contributes to ER⁻ breast cancer primary tumors metastasis to the lung. This suppression enables adhesion to the lung parenchyma, thus facilitating seeding at the lungs. Moreover, RARRES3 suppression and loss of its PLA$_{1/2}$ catalytic activity disable differentiation signals, which, in turn, provide metastasis initiation competence to breast cancer cells to colonize the lung.

    

van Lohuizen, 2006). GATA3 has been shown to suppress lung metastasis from mouse and human mammary tumors by a mechanism that involves cell fate specification (Kouros-Mehr *et al*, 2008a; Dydensborg *et al*, 2009). In contrast, the maintenance of a stem phenotype, associated with self-renewal properties, is critical for cells to establish new lesions (Al-Hajj *et al*, 2003; Dontu *et al*, 2003; Li *et al*, 2007; Liao *et al*, 2007), which in ER⁻ BC would be favored by the loss of expression or catalytic activity of RARRES3.

Our data suggest that the $PLA_{1/2}$ catalytic activity of RARRES3 is central to its differentiation function. $PLA_{1/2}$ enzymes are catalytically responsible for the production of arachidonic acid, which is subsequently processed to produce prostanoids and leukotrienes (Wang & Dubois, 2010). Prostaglandins and leukotrienes modulate the proliferation, migration, and invasion of tumor epithelial cells through multiple signaling pathways in both an autocrine and paracrine fashion (Wang & Dubois, 2010). Moreover, these signal mediators are key molecules in the regulation of differentiation and stem cell homeostasis (Ginestier *et al*, 2009; Wang & Dubois, 2010). Here we show that the strong requirement of the $PLA_{1/2}$ catalytic activity of RARRES3 to sustain differentiation reflects, in part, an increase in PPAR activity and expression of its downstream targets. PPAR signaling provides a survival advantage to BC cells upon loss of attachment (Carracedo *et al*, 2012), and activation of PPARA with the chemical agonist compound Wy14643 reduces the development of malignant mammary tumors in a tumor-prevention setting (Pighetti *et al*, 2001). Some of the PPAR-regulated pro-differentiation activities described herein may be reduced in BC cells expressing low levels of RARRES3, thus facilitating metastatic features. This may explain why BC primary tumors that will metastasize express low levels of *RARRES3*. The induction of differentiation appears to be a common mechanism by which cells restrain their metastatic capacity in tumors of distinct origin. However, this observation does not explain why, among ER⁻ tumors, those expressing low levels of *RARRES3* exhibit significantly poorer lung metastasis-free survival, while metastasis to bone or brain remains largely unaffected.

In addition to supporting a reduction in metastasis initiation capacity, we show that RARRES3 modulates the ability of metastatic tumor cells to specifically attach to the lung parenchyma, which may explain the specific differences observed in lung extravasation capacity. *RARRES3* loss of expression favored the adhesion of ER⁻ BC cells to extracellular matrix proteins of the lung. While the lung parenchyma is composed mainly of Types I and III collagen, elastin, fibronectin, proteoglycans, and glycosaminoglycans (Suki *et al*, 2005; Pelosi & Rocco, 2008), the brain extracellular matrix has a unique composition, and matrix proteins common in other tissues are virtually absent in the brain (Dityatev *et al*, 2010). The lack of association of *RARRES3* expression levels with brain metastasis may be explained by the fact that *RARRES3* specifically modulates adhesion to the lung parenchyma and not to that of the brain. In addition to extravasation and homing through an endothelial cellular layer and to a specific matrix, the blood–brain barrier (BBB) may be a limiting step for cell colonization of the brain (Cardoso *et al*, 2010). Thus, extravasation to the brain may require the concerted acquisition and loss of expression of multiple genes, while a single gene would not have sufficient strength to drive this phenotype (Bos *et al*, 2009). On the other hand, the absence of a vasculature barrier or collagen/fibronectin-rich matrix to overcome in bone metastasis might be the molecular rationale for the lack of *RARRES3* downregulation in highly metastatic

populations to the bone (Kang *et al*, 2003). In summary, the decrease in *RARRES3* expression may confer metastatic cells an advantage to adhere to the lung parenchyma, thus facilitating subsequent lung colonization over other potential metastatic tissues.

Our data indicate that *RARRES3* is a clinically relevant gene that restrains the lung metastatic capacity of BC cells and whose levels in the primary tumor may also predict risk of specific relapse. The contribution of RARRES3 to differentiation over self-renewal suggests that reduced *RARRES3* expression would also be predictive for cancer patients that exhibit therapy-resistant tumors. In fact, the characteristics that stem cells exhibit underlie their capacity to survive conventional therapies (Schott *et al*, 2013). Therefore, tumors expressing low levels of *RARRES3* may require new therapies designed to target BC-initiating cells. In addition, our results support the notion that RARRES3 activation leads to the differentiation of BC tumor cells and contributes to limiting metastasis progression. Thus, screening for compounds that activate RARRES3 may contribute to the development of new differentiation-inducing strategies to target therapy-resistant tumors. Alternatively, depending on the latter strategies effectiveness to enforce differentiation, RARRES3 activation could offer a useful pretreatment to improve the effect of conventional therapies. On the basis of the mechanistic and clinical data presented above, it is suggestive to use retinoic acid in the adjuvant setting to induce RARRES3 metastasis suppressor function, given its current use to treat certain hematological diseases. The chemopreventive use of retinoids has been described to reduce the appearance of secondary neoplasias in patients with lung, head and neck, liver, and breast cancer (Fields *et al*, 2007). Nevertheless, some contradictory results have strongly curtailed its potential in the treatment of solid tumors (Lotan *et al*, 1995) and may support the development of alternative strategies to increase *RARRES3* expression.

## Materials and Methods

### Cell culture

LM2 cell derivative is a lung metastatic subline derived from the MDA-MB-231 breast cancer cell line in Prof. Massagué's laboratory (Minn *et al*, 2005). CN37 is a pleural effusion patient-derived cellular population (Gomis *et al*, 2006). 4T1 cells were originated from a spontaneous BALB/c mouse breast cancer tumor (Aslakson & Miller, 1992). Stable cell lines expressing the shRNA RARRES3 or a non-silencing shRNA were generated as described (Tarragona *et al*, 2012). For RARRES3 overexpression in cells, the RARRES3 sequence was cloned into the retroviral vector pBabePuro/hygro. Stable cell lines expressing the various vectors described were generated under puromycin selection for 48 h or hygromycin selection for 14 days. All cell lines were stably transfected with TK-GFP-Luciferase construct and sorted for GFP.

### Animal studies and xenografts/syngeneic models

All animal work was approved by the institutional animal care and use committee of IRB Barcelona. Female BALB/c Nude (MDA-MB-231 cells), NOD/SCID (CN37 cells), or BALB/c wild-type mice (4T1 cells) were used.

For tail vein injections, cells were resuspended in 1× PBS and injected into tail vein of mice using a 26G needle, as previously described (Tarragona et al, 2012). Prior to the injection of tumor cells, mice were anesthetized with ketamine (100 mg/kg body weight) and xylazine (10 mg/kg body weight), and immediately after injection they were imaged for luciferase activity. Mice were monitored weekly using IVIS imaging, unless otherwise indicated. Lung tumor development was followed once a week by biolumines-cence imaging, taking a photo of upper dorsal region that corre-sponds to lung position. Bioluminescent images were quantified with Living Image 2.60.1 software. All obtained values were normal-ized to those obtained at day 0. 4T1 lung colonization capacity was scored 20 days post-inoculation by H&E. Three sections per mouse lungs separated 25 μm were counted. The average of the total area of the metastasis normalized to total lung area was measured. Then, the average of all mice total lung metastasis area was plotted.

For the injection of tumor cells at the orthotopic site, mice were anesthetized as described above, and tumor cells mixed with growth factor-reduced matrigel (BD Bioscience) before inoculation (1:1). Once palpable, tumors were measured with a digital caliper, and the tumor volume was calculated. For metastatic experiments in Fig 2A, tumors were resected when reaching 300 mm$^3$.

For the injection of cells directly into the lungs, mice were anes-thetized as described above, and the indicated number of cells was counted and then suspended in 25 μl of 1× PBS and mixed 1:1 with growth factor-reduced matrigel (BD Bioscience). To avoid injecting the heart of BALB/c Nude, NOD/SCID, or BALB/c wild-type mice, a total of 50 μl of this solution was injected directly between the 3rd and the 4th costal bone. Dilutions including 50, 500, 1,000, and 5,000 cells were used. On the day of injection (day 0), luciferase activity was assessed with IVIS. Subsequently, this activity was measured to score tumor initiation of colonization. In case of 4T1 cells, 20 days post–injection, mice were killed and lungs analyzed for macroscopic lesion detection. Lesions were confirmed by H&E staining.

For in vivo lung extravasation assays, CellTracker™ Green (Invi-trogen)-marked cells (5 × 10$^5$) were suspended in 200 μl of cold 1× PBS and injected into the tail vein. After 48 h, we then injected 50 μg (100 μl) of rhodamine-lectin into the same vein to label the vasculature. Mice were perfused via heart with 5 ml of 1× PBS and sacrificed 30 min later. Lungs were removed, the trachea was perfused, and lungs frozen in OCT. OCT sections were then analyzed.

For in vivo tumor permeability assays, mice were injected intra-venously with rhodamine-dextran (70 kDa, Invitrogen) at 2 mg per 20 g of body weight, and 3 h later they were perfused via heart with 5 ml of PBS and sacrificed. Tumors were extracted and fixed in formalin. Paraffin-embedded tumors were sectioned and analyzed.

**Oligonucleotide array assays**

RNA sample collection and generation of biotinylated complemen-tary RNA (cRNA) probe were carried out essentially as described in the standard Affymetrix (Santa Clara, CA) GeneChip protocol. Ten micrograms of total RNA was used to prepare a cRNA probe using a Custom Superscript kit (Invitrogen). For expression profiling, 25 ng of RNA per sample was processed using isothermal amplifica-tion SPIA Biotin System (NuGEN technologies). Each sample was

hybridized with an Affymetrix Human Genome U133APlus2.0 microarray at the IRB Barcelona Functional Genomics Facility. All microarray statistical analyses were performed using Bioconductor (Gentleman et al, 2004). Background correction, quantile normali-zation, and RMA summarization were performed as implemented in Bioconductor's affy package (Irizarry et al, 2009).

**Patient gene expression datasets**

The patients' information is publically available and was down-loaded from GEO Barrett et al (2007). The following cohorts were used: (i) MSKCC set. GSE2603, including 82 breast cancer samples (Minn et al, 2005); (ii) MSK/EMC. Pooled GSE2603, GSE2034, GSE5327, and GSE12276. This pooled cohort has 560 patients' samples. In order to remove systematic biases, the expression measurements were converted to z-scores for all genes prior to merging. ER$^+$ patients were selected based on the bimodality of gene ESR1. More information is provided in Supportive materials and methods.

**Lentiviral and retroviral production**

293T cells were used for lentiviral production. Lentiviral vectors expressing shRNAs against human RARRES3 from the Mission shRNA Library were purchased from Sigma-Aldrich. Cells were transfected with lentiviral vectors following standard procedures, and viral supernatant was used to infect MDA-MB-231 and CN37 cells. Selection was done using Puromycin (2 μg/ml) for 48 h. As a negative control in all the infections, a lentivirus with control shRNA was used. 293T cells transfection with retroviral vectors was done using standard procedures, and viral supernatant was used for infection. An empty vector was used as a Mock control.

RARRES3 short hairpins sequence:
sh#1: CCGGCCCGCTGTAAACAGGTGGAAACTCGAGTTTCCACCT-GTTTACAGCGGGTTTTTG
sh#2: CCGGGCGCTTGGAATCCTGGTTGTTCTCGAGAACAACCAG-GATTCCAAGCGCTTTTTG
Control short hairpin:
ShControl: CCGGCATCGACAAGACTGCTAACCACTCGAGTGGTT-AGCAGTCTTGTCGATGTTTTTG

**Statistical analysis**

Metastasis-free survival curves of mice were plotted using Kaplan–Meier estimates and compared using the log-rank test. Categorical variables were compared with the Fisher exact test. Continuous variables were compared nonparametrically with the Wilcoxon test or with a Student t-test depending on normality of the distribution. Irrespectively of whether the direction of the differences was biolog-ically expected to follow a certain direction (i.e. gene silencing), two-sided tests were used, unless indicated otherwise. We consid-ered $P < 0.05$ to be statistically significant.

Kaplan–Meier survival and correlation analysis in patient samples: Publicly available and clinically annotated breast cancer cohorts with gene expression profiles (GSE2603, GSE2034, GSE5327, and GSE12276) were pooled as described above. Various probes of the same gene were summarized via mean. Patients were divided into groups on the basis of levels of expression using natural

divisions (i.e. tertiles, median), and the Kaplan–Meier survival function was plotted. The hazard ratio (HR) and *P* value for each gene of interest (RARRES3 or ESR1) were calculated using a Cox proportional hazards model and performing likelihood ratio tests. The HR was checked for constancy over time, fulfilling Cox model assumptions. All significance measurements were done using the parameter of interest, RARRES3 or ESR1, expression as a continuous variable.

### GSEA analysis

GSEA analysis was done as implemented in the phenoTest package of Bioconductor.

### RARRES3 gene expression correlation

Gene expression data of ER⁻ patients ($n = 211$) of GSE2603, GSE2034, GSE5327, and GSE12276 pooled breast cancer sample cohort were used. A Spearman correlation test was performed for each gene against RARRES3. We corrected for multiple testing using the Benjamini and Hochberg method.

### Protein extraction and Western blot

Cells were lysed with a buffer containing 1% Triton in 50 mM Tris/HCl (pH 7.4) for protein extracts and processed as in Tarragona *et al*, 2012. The antibodies used were anti-RARRES3 (Abyntek SA) and α-Tubulin (Sigma). RARRES3 rabbit polyclonal antibody was generated using RARRES3 (23-117Aa) produced in *E. coli*.

### Quantitative real-time PCR

Total RNA was isolated and processed as described (Tarragona *et al*, 2012). Human *RARRES3,* the other genes described (i.e. *PPARG, ACSL5,* and *ID1*), human *B2M* and mouse *B2M or GAPDH* as endogenous controls were amplified with commercially designed TaqMan gene expression assays (Applied Biosystems).

### RNA isolation from metastasis tumors

Lungs positive for luciferase observed *ex vivo* were collected, and RNA was obtained by adding 600 µl of lysis buffer (from Ambion kit) plus 1% β-Mercaptoethanol directly to the lungs. Lung tissue was homogenized using a Pre-cellys 24 machine (20 s, two cycles) (Bertin Tech). The homogenized extract was then passed through a QIAshredder column (Qiagen, cat no. 19656), and RNA was purified using a PureLink RNA mini kit (Ambion, cat no. 12183918A), following the manufacturer's instructions.

### Histopathology and immunohistochemistry

Tissues were dissected, fixed in 10% buffered formalin (Sigma), and embedded in paraffin or fixed-frozen OCT. Sections (2- to 3-µm thick) were stained with hematoxylin and eosin (H&E). For Ki67 and Vimentin IHC staining, paraffin sections were deparaffinized and rehydrated through a series of alcohols. Next, sections were treated with peroxidase-blocking solution for 15 min and washed two times with distilled water. In particular, for Ki67 IHC antigen retrieval, sections were boiled for 20 min in citrate buffer pH6. They

were then washed three times with 1× PBS and blocked with 0.05% BSA in 1× PBS for 30 min at room temperature. Then, sections were incubated with primary antibody against human Ki67 (Novocastra NCL-ki67p; dilution 1:500 in 0.05% BSA, 1× PBS) for 1 h at room temperature. They were then washed three times with 1xPBS and incubated with HRP-conjugated secondary antibody raised against rabbit IgGs (BrightVision poly HRP-Anti_Rabbit IgG ready to use; ImmunoLogic) for 45 min at room temperature. Slides were washed three times with 1× PBS and incubated with DAB for 3 min. Hematoxylin counterstaining was then performed.

For Vimentin IHC antigen retrieval, sections were autoclaved for 10 min in citrate buffer (pH 6.0). Next, sections were washed three times with 1× PBS and blocked with 1× PBS for 30 min at room temperature. They were then incubated with primary antibody against human Vimentin (Novocastra NCL-L-VIM-V9; dilution 1:100 in 1× PBS) for 2 h at room temperature. Further, sections were washed three times with 1× PBS and incubated with HRP-conjugated secondary antibody raised against mouse IgGs (Bright Vision poly HRP-Anti Mouse IgG ready to use; ImmunoLogic) for 30 min at room temperature. Slides were then washed three times with 1× PBS and incubated with DAB for 3 min. Hematoxylin counterstaining was then performed.

For PPARD IHC antigen retrieval, sections were incubated in citric buffer (pH 6.0) at 95°C 30 min. Mouse monoclonal PPARD F-7 antibody (Santa Cruz Biotechnology SC-74440) was used in 1:15 dilution. IHC detection was performed with the ABC kit, from Vector Laboratories. Slides were counterstained in Harris hematoxylin, dehydrated, cleared, and cover-slipped.

For quantification of the number of foci per field, images from Vimentin-immunostained lung sections were taken at 2× magnification (three sections per lung and five animals per group). For each section, the average number of foci per field was plotted. To analyze the metastatic area, images were taken at 4× magnification, and the area of each metastatic lesion was quantified with the Image J software. Five images per section/animal were evaluated, and the average area plotted.

For quantification of Ki67, images from Ki67-immunostained lung sections were taken at 40× magnifications (five fields per section and five sections per lung lesion). Percentage of Ki67-positive cells relative to total number of cells was quantified. Total of five mice per each group were analyzed.

### Reporter assays

Renilla and luciferase reporter assays were performed as previously described (Tarragona *et al*, 2012). The plasmid 3xAOX PPRE-TK-LUC containing three copies of the peroxisome proliferator-response element (PPRE) from the rat acyl-CoA oxidase was used. A Renilla plasmid (Promega) was included to control for transfection efficiency.

### Migration assay

Cells were marked with 5 µM CellTracker™ Green (Invitrogen) following the manufacturer's instructions and kept overnight in medium with 0.1% FBS. Next day, $5 \times 10^4$ cells were seeded on human fibronectin-coated Biocoat Cell Culture Inserts (Becton Dickinson Labware) in medium with 0.1% FBS, while the wells were

loaded with complete medium. Eight hours after seeding, cells were washed and fixed with 4% paraformaldehyde. Cells on the apical side of each insert were scraped off, and migration to the basolateral side was visualized with Nikon Eclipse TE2000-U fluorescent microscope. Each sample was seeded in triplicate, and five fields from each well were counted.

**Invasion assay**

Cells were marked with 5 μM CellTracker™ Green (Invitrogen) following the manufacturer's instructions and were kept overnight in medium with 0.1% FBS. Next day, $5 \times 10^4$ cells were seeded on chambers coated with growth factor-reduced or completed matrigel (BD Bioscience) in medium with 0.1% FBS, while the wells were loaded with complete medium. Eight hours after the seeding, cells were washed and fixed with 4% paraformaldehyde. Cells on the apical side of each insert were scraped off, and the migration to the basolateral side was visualized with Nikon Eclipse TE2000-U fluorescent microscope. Each sample was seeded in triplicate, and five fields from each well were counted.

**Adhesion assay**

Cells were marked with 5 μM CellTracker™ Green (Invitrogen) following the manufacturer's instructions and kept overnight in medium with 0.1% FBS. Next day, $5 \times 10^4$ cells were seeded in triplicates on collagen- or fibronectin-coated 24-well inserts. One hour after the seeding, cells were washed and fixed with 4% paraformaldehyde. They were then visualized with a Nikon Eclipse TE2000-U fluorescent microscope. Each sample was seeded in triplicate, and five fields from each well were counted.

**Flow cytometry analysis**

Cells were stained using the Annexin V Apoptosis Detection kit (BD Pharmingen), following manufacturer's instructions. Data were obtained using a BD FACSAria cell sorter and analyzed using FlowJo software.

**Oncospheres formation assay**

To assess tumor initiation capacity *in vitro*, cells were counted and plated into low-attachment 96-well plates at dilution of 1 cell per well. They were then cultured in mammary epithelial basal medium (MEBM, Lonza, cat no. CC-3151), supplemented with MEGM Single-Quots (which contain Insulin, EGF, Hydrocortisone and GA-1000, LONZA cat no. 4136), 1X B27 without retinoic acid (GIBCO, cat no. 12787-010), and 20 ng/ml of recombinant fibroblast growth factor (GIBCO, cat no. PHG0026), and incubated in 5% CO2, 37°C in order to obtain a first generation of oncospheres (anoikis and pluripotency selection) after 15 days. The process was repeated to ensure second-generation oncospheres (pluripotency selection). After 2 weeks of culture, the oncospheres were counted under the microscope.

**Organotypic 3D formation assay**

Cells from first-generation oncospheres were spun down at 100 ×g for 5 min. The pellet was then disaggregated using 0.5% trypsin (Sigma, cat no. T-3924) for 5 min at 37°C. Trypsin was blocked using DMEM/F12 medium (GIBCO) supplemented with 10% FBS, and subsequently cells were spun down at 600 ×g. Cells were counted and then resuspended in growth-reduced factor matrigel (BD Bioscience, cat no. 354230) in order to obtain 1,000 cells per 50 μl. Each drop was placed in the center of one well of an adherent 24-well plate and incubated for 15 min. After gel solidification, each well was replenished with 400 μl of MEBM medium supplemented with the same factors described in the oncosphere formation assay. Media was replaced every 2 days, and organotypic 3D structures were grown for 15 days. Total spheroids were counted in each drop and considered positive when exceeding 50 cells and a diameter of 50 μm.

**Arachidonic acid levels determination**

To determine arachidonic acid in cell extracts, a total of 10 million cells were collected in 1 ml of 1× PBS and stored at −20°C. In order to break the membranes, two cycles of freeze-thawing were performed. Cells were then spun down at 2,000 ×g for 5 min at 4°C, and supernatant was collected. Fifty microliters of each condition was dispensed into the human arachidonic acid (AA) ELISA kit (cat. no CSB-E09040 h, CUSABIO). Each condition was assessed in triplicate, and standard curve and concentrations were assessed using the professional soft "Curve Expert 1.3" provided by CUSABIO. The data plotted are the average of three independent experiments.

**Circulating tumor cells**

Blood from mice was collected in tubes containing EDTA/heparin. The fluid was transferred to 2-ml plastic tubes and centrifuged for 10 min at 86 ×g at 4°C. The supernatant was discarded. If the pellet was bloody, 1 ml of ACK lysis buffer (Cambrex: 10-548) was added for 5 min at room temperature and after that, the collected sample was mixed with PBS to a total volume of 10 ml, centrifuged again and decanted. RNA of the remaining cells was extracted. Human B2M and mouse GAPDH taqman probes were used to assess the amount of human versus mouse RNA in mouse blood.

**Supplementary information** for this article is available online: http://embomolmed.embopress.org

## Acknowledgements

We would like to thank the Functional Genomics, Microscopy, and Cytometry core facilities of IRB Barcelona, and the UB. We thank C. Caelles for the 3AOX-luc construct. We thank Angel Nebreda for his scientific suggestions. EJA is supported by "La Caixa" PhD fellowship programme, and JU is a Juan de la Cierva Researcher (MICINN). JM is a Howard Hughes investigator. The work of A.C. and S.F-R is supported by the Ramón y Cajal award to AC (Spanish Ministry of Education) and the ERC (336343). JM was supported by HHMI. RRG and XS are ICREA Research Professors (Institució Catalana de Recerca i Estudis Avançats). Support and structural funds were provided by the *Associación Española Contra el Cáncer* (AECC), Fundación BBVA, *Generalitat de Catalunya* (2009 SGR 1429), and Spanish *Ministerio de Ciencia e Innovación* (MICINN) (SAF2010-21171) to RRG.

## Author contributions

MM and EJA designed and performed experiments, analyzed data, and wrote the paper. JU performed and analyzed experiments and wrote the paper. EF

## The paper explained

### Problem

Breast cancer is the most frequently diagnosed cancer in women in Europe and the United States. Despite a recent decrease in the incidence of this disease, it continues to be the second leading cause of death by cancer. Most of these deaths are caused by the metastatic spread of the tumor. The lung is a common site of metastatic relapse in ER-negative breast cancer patients, and metastasis is responsible for the fatal outcome of the disease. Thus, a better understanding of the biology of the metastatic process is needed if we are to tackle this problem.

### Results

In this study, we show that *RARRES3* is a metastatic suppressor gene in breast cancer. Using the MDA-MB-231 breast cancer cell line model and derivatives, which have a strong metastatic capacity to lung, we functionally validated that RARRES3 loss of expression in ER-negative breast cancer cells confers a selective advantage for the colonization of the lung. Tumor cells sometimes cannot grow or survive in the absence of a supportive microenvironment. We show that loss of *RARRES3* expression facilitates the ability of the tumor cells to extravasate and adhere to the lung extracellular matrix and facilitates the initiation of proliferation to colonize the lung. Collectively, our results show that genes selected for metastasis contribute to the different steps of this process and represent the random accumulation of traits that provide the necessary advantage for adaptation to the microenvironment of a different organ.

### Impact

This study shows that *RARRES3* restrains the lung metastatic capacity of breast cancer cells and that *RARRES3* levels in the primary tumor are clinically relevant as may predict risk of relapse. The contribution of RARRES3 to differentiation over self-renewal suggests that reduced *RARRES3* expression could be also predictive of therapy-resistant tumors, identifying patients possibly requiring new therapies designed to target breast cancer-initiating cells. Thus, screening for compounds that activate RARRES3 may contribute to the development of new differentiation-inducing strategies to target therapy-resistant breast tumors.

and MG performed experiments with cells, IHC, and all experiments with mice, together with MM. SF-R performed IHC experiments. EP performed all the bioinformatics experiments. RBF and XS generated and analyzed the 3D structural model of RARRES3. DR produced RARRES3 fragment in *E. coli*. AC analyzed data and design experiments. JM designed experiments and analyzed data. RRG supervised the project, designed experiments, analyzed data, and wrote the paper.

## Conflict of interest

The authors declare that they have no conflict of interest.

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
