## [Review Process File · EMBO Molecular Medicine]

RARRES3 suppresses breast cancer lung metastasis by regulating adhesion and differentiation

Mònica Morales, Enrique J. Arenas, Jelena Urosevic, Marc Guiu, Esther Fernández, Evarist Planet, R. Bryn Fenwick, Sonia Fernández-Ruiz, Xavier Salvatella, David Reverter, Arkaitz Carracedo, Joan Massagué, Roger R. Gomis

Corresponding author: Roger Gomis, Institute For Research in Biomedicine

Review timeline:

Submission date:	18 November 2013
Editorial Decision:	17 December 2013
Revision received:	03 April 2014
Accepted:	23 April 2014

Transaction Report:

Editor: Roberto Buccione

1st Editorial Decision

17 December 2013

Thank you for the submission of your manuscript to EMBO Molecular Medicine. We have now heard back from the three Reviewers whom we asked to evaluate your manuscript.

You will see that while the Reviewers are globally supportive of your work, they express a number of concerns that prevent us from considering publication at this time. I will not dwell into much detail, as the evaluations are detailed and self-explanatory and I will just mention a few main points.

Reviewer 1 is more critical and points to significant issues that require your action, and that mainly deal with insufficient quality of the data and limited experimental support of some claims. I believe the criticisms to be quite detailed and thorough. This Reviewer also raises doubts as to the fact that direct inoculation of tumor cells into the lung is a true assay of reinitiation. Finally, s/he notes that the quality of presentation is insufficient and needs substantial improvement. I agree with this assessment.

Reviewer 2 has a significant concern, with which we agree. S/he finds the study to be limited by the use of only two cell lines and would like to see further experimentation in a suitable model. We agree that this would considerably improve the manuscript and make it more suitable for EMBO Molecular Medicine and thus encourage you to provide such experimentation. The Reviewer also notes that a connection is not clearly made between how RARRES3 and MTDH can bind and that altering RARRES3 changes adhesion of cells to collagen I or fibronectin. Reviewer 2 also lists a few other items that require your attention.

Reviewer 3 also notes the weakness deriving from the use of cell lines and the insufficient quality of data presentation and description.

Considered all the above, while publication of the paper cannot be considered at this stage, we would be prepared to consider a substantially revised submission, with the understanding that the Reviewers' concerns must be fully addressed, with additional experimental data where appropriate and that acceptance of the manuscript will entail a second round of review.

Please note that it is EMBO Molecular Medicine policy to allow a single round of revision only and that, therefore, acceptance or rejection of the manuscript will depend on the completeness of your responses included in the next, final version of the manuscript.

As you know, EMBO Molecular Medicine has a "scooping protection" policy, whereby similar findings that are published by others during review or revision are not a criterion for rejection. However, I do ask you to get in touch with us after three months if you have not completed your revision, to update us on the status. Please also contact us as soon as possible if similar work is published elsewhere.

I look forward to seeing a revised form of your manuscript as soon as possible.

***** Reviewer's comments *****

Referee #1 (Comments on Novelty/Model System):

I am not sure that direct inoculation of tumor cells into the lung is an assay of reinitiation

Referee #1 (Remarks):

In this manuscript, Morales et al. have investigated the role of the putative suppressor gene RARRES3 in breast cancer metastasis to the lung. RARRES3 was originally identified by the Massague lab as a downregulated gene within their "lung metastasis signature". Here the function and mechanisms of action of RARRES3 in lung metastases is investigated. The data presented represents a very large body of work (many in vivo experiments, proteomics, biochemical analysis, structural modeling, in vitro cell biology etc.). The results obtained are of certainly of interest and this research area is suitable for EMBO Mol Medicine.

However, the major issue with this manuscript is that it is very poorly written and presented. Many parts of this manuscript look like they have been cut and pasted from a previous, differently formatted version and/or that this is a merge of different projects/manuscripts. Data are presented but then not followed up. Overall this makes for a difficult read and, in parts and difficult to make a proper judgment of the data and findings. On this basis alone I would recommend a major revision before being properly considered.

Examples of the problems with this manuscript

1: Insufficient details/errors

(a) All materials and methods to be rewritten. For example

(i) some methods are missing (circulating tumor cells, intrapulmonary injections, Ki67 antibody information and staining protocols, culture conditions for oncosphere etc). Information on the non-

targeting sh-control (NTC) is missing

(ii) some methods are described but not used (e.g. intracardiac injections)

(iii) some numbers do not match between the figures, text and M&Ms (e.g. page 5 MSKCC primary breast cancer set in text n=86 versus n=84 in the M&M. Figure 6a: 500 in the Figure versus 10,000 cells in M&Ms)

(v) what do the author mean by "control vector parental" - see Fig 3b)

(vi) details as to how foci and tumor area in the lung was quantified

(vii) The authors state, that they used FFPE sections according to the M&Ms, in the Figure legend 4B they state that they used frozen sections. The authors refer to HA-RARRES3 in the text, when it should be FLAG-RARRES3 (and many other similar typographical issues)

(b) statistical analyses are poorly described or missing

(c) references that don't match the text (e.g. page 10, text describing how a poor prognosis suppressor gene might function to impair adhesion to the lung parenchyma followed by 3 references on breast cancer stem cells).

(d) many grammatical issues and just unclear statements

2: The authors present a very confusing introduction and an abstract that doesn't fully support the authors' claims e.g.

(a) What is the evidence that "RARRES3 is negatively selected in tumors with high risk of lung metastasis?" what is the evidence that this is selection for RARRES3 rather than for subpopulations of cells?

(b): What is the evidence that RARRES3 suppression facilitates reinitiation abilities at the metastatic site? The authors provide data that if the cells are inoculated directly into the lung that RARRES3 expression impairs the level of tumour burden in the lung but this is not an assay for reinitiation

(c): How can RARRES3 be a target - you wouldn't want to target a metastasis suppressor.

3: The authors show that RARRES3 can interact with MTDH (Figure 4E,F) but then never follow this up. Either they should demonstrate that this interaction is functionally important or remove this data.

4: Why did the authors investigate PPAR target genes?? Again, not clear where this part of the manuscript is going.

5: A key conclusion from this manuscript is that RARRES3 expression "enforces breast cancer tumors to preserve their differentiation attributes". I may have been so muddled by this stage but it is not clear to me that the authors have been monitoring differentiation in their tumor cells nor the role of PPAR in this process.

6: Figure 2B. The authors state that they remove the tumors after 28 days, when the tumors had approximately 300 mm³. According to their growth curves in 2B the tumors had a size of approximately 100-150 mm³

7: The authors show a very strong in vivo and ex vivo IVIS images for the LM2 lungs in Figure 2C. However, in the IHC vimentin staining in Figure 2D only single cells are found. According to the IVIS picture, you would expect proper mets and clusters in the lungs. The authors should show matched IVIS and IHC images from the same set of lungs. i.e. the images shown do not seem to be representative. In addition, higher power images of Figure 2D would be helpful

8: As far as I can make out, the authors compare LM2 cells versus LM2-RARRES3 cells. The control that is needed is LM2 cells transfected with empty vector.

9: The authors show that LM2 and LM2-RARRES3 cells injected into the MFP give rise to primary tumors with similar growth rates. Is the same true for the CN37a and CN37-shRARRES3 cells?

10: Figure 4B. It is not possible to determine whether the cells have extravasated. The authors have flushed/perfused the lungs, but they should include vessel markers to exclude cells remained adhered to the inside of vessels.. What is the red stain - background???

11. What do the red and green colors in Figure 5D represent. How were the oncospheres evaluated in Figure 6B?

Referee #2 (Comments on Novelty/Model System):

Most of the studies relied on two cell lines, and it would be nice to use PDX's and/or syngeneic GEM models with an intact immune system as well

Referee #2 (Remarks):

Morales et al. clarify the role/molecular function of RARRES3, a previously identified lung metastasis suppressor gene, during pulmonary metastasis. The authors predominantly utilize 231 and LM2 cells to demonstrate that reduced RARRES3 levels promote pulmonary metastases. The authors propose and their experiments support a dual mechanism whereby RARRES3 blocks metastasis. First, it binds to Metadherin thereby preventing cancer cell adhesion to the lung. Second, it promotes a dedifferentiated state that is conducive to metastatic outgrowth/reinitiation. Overall, the experiments performed are well executed and controlled and, in general, the authors' conclusions are supported by their data.

Major points/Questions:

The authors show that RARRES3 and MTDH can bind and that altering RARRES3 changes adhesion of cells to collagen I or fibronectin, but a strong connection of the two events is not made. For example, does MTDH overexpression rescue RARRES3-expression induced block in adhesion? How does the physical interaction of RARRES3 with MTDH disrupt its function during adhesion? Please expand on in the Discussion or provide experiments to elucidate the mechanism.

Minor points:

Section title for second section of results currently reads "RARRES3 Promotes Breast Cancer Lung Metastasis". Perhaps this is a typo and should read "RARRES3 depletion promotes breast cancer lung metastasis" or "RARRES prevents breast cancer lung metastasis".

Please show complete dual sided error bars for all bar graphs (e.g. Fig. 4C, D)

While the authors do confirm that Aox-3x-PPRE-Luc reporter is responsive to RARRES3 catalytic activity, to claim that "PPAR transcriptional activity could be used as a surrogate marker of RARRES3 PLA1/2 activity" appears to be a bit of an overstatement. Many other cellular events may contribute to the final readout of the luciferase reporter.

Are arachidonic acid levels, prostinoids and/or leukotrienes reduced in the regions of downregulation of RARRES3 in human tumors or in vitro models? Does this correlate with outcome?

Referee #3 (Comments on Novelty/Model System):

The model is based on a single cell line, which is generally a weakness, but the data were substantiated by analysis of human tumors

Referee #3 (Remarks):

The manuscript by Morales et al. investigates the contribution of RARRES3 in breast cancer metastases to the lung. Based on the observation RARRES3 is downregulated in ER--ve breast cancer primary tumors that will metastasize, the authors build the case that RARRES3 acts at two levels: first to impede attachment to the tissue-specific fibronectin/collagen ECMs present in lung, and second to promote retention of differentiation characteristics that block reinitiation of cancer growth in the lung. The latter is shown to be dependent of PPAR activity. The authors suggest that RARRES3 status might be useful prognostically in breast cancer, while acknowledging it is specific to lung metastatic risk.

These conclusions are generally carefully documented with specific consideration of alternative mechanisms. The paper is well-written overall but there are gaps in the data follow through that might be addressed, such as the importance of MTDH for phenotype. There are limiting weakness is the description of methods and lack of sufficient information about the size and replication of experiments, particularly cellular and in vitro data, on which conclusions are based. For example whether N= mice or fields or slides in Figure 3D. This information should be explicitly stated in figures, text or figure legends, including number of experiments for which a representative experiment is shown.

A confusing point is made on Pg 7, and again in Figure 3 C: the statement "CN37 patient-derived ER metastatic breast cancer cells' is a bit odd as it implies a primary or cells strain, i.e. recently patient-derived, but the sentence goes on to compare these cells to 'parental' well-established cancer cell line, MDA-MB-231. Please clarify.

Point by point response to the reviewers' comments of manuscript (EMM-2013-03675) now entitled "***RARRES3 suppresses breast cancer lung metastasis by regulating adhesion and differentiation***".

Reviewer #1

General Statement. "...In this manuscript, Morales et al. have investigated the role of the putative suppressor gene *RARRES3* in breast cancer metastasis to the lung. *RARRES3* was originally identified by the Massague lab as a downregulated gene within their "lung metastasis signature". Here the function and mechanisms of action of *RARRES3* in lung metastases is investigated. The data presented represents a very large body of work (many in vivo experiments, proteomics, biochemical analysis, structural modeling, in vitro cell biology etc.). The results obtained are of certainly of interest and this research area is suitable for *EMBO Mol Medicine*. However, the major issue with this manuscript is that it is very poorly written and presented. Many parts of this manuscript look like they have been cut and pasted from a previous, differently formatted version and/or that this is a merge of different projects/manuscripts. Data are presented but then not followed up. Overall this makes for a difficult read and, in parts and difficult to make a proper judgment of the data and findings."

Reply: We appreciated the referee's statement regarding the interest and the suitability of our research for *EMBO Mol Medicine*. We acknowledge the limitations of the previously submitted version. Below we clarify how we have addressed the concerns raised.

Point 1: Insufficient details/errors

(a) All materials and methods to be rewritten. For example

- (i) some methods are missing (circulating tumor cells, intrapulmonary injections, Ki67 antibody information and staining protocols, culture conditions for oncosphere etc). Information on the non-targeting sh-control (NTC) is missing
- (ii) some methods are described but not used (e.g. intracardiac injections)
- (iii) some numbers do not match between the figures, text and M&Ms (e.g. page 5 MSKCC primary breast cancer set in text $n=86$ versus $n=84$ in the M&M. Figure 6a: 500 in the Figure versus 10,000 cells in M&Ms)
- (v) what do the author mean by "control vector parental" - see Fig 3b)
- (vi) details as to how foci and tumor area in the lung was quantified
- (vii) The authors state, that they used FFPE sections according to the M&Ms, in the Figure legend 4B they state that they used frozen sections. The authors refer to HA-*RARRES3* in the text, when it should be FLAG-*RARRES3* (and many other similar typographical issues)

(b) statistical analyses are poorly described or missing

(c) references that don't match the text (e.g. page 10, text describing how a poor prognosis suppressor gene might function to impair adhesion to the lung parenchyma followed by 3 references on breast cancer stem cells).

(d) many grammatical issues and just unclear statements

Reply: Following these suggestions, we have carefully revised the manuscript to address all the points listed above and others that we have further encountered. In summary, we have modified

the text to streamline the message, to provide format consistency, and to detail material and methods. In particular, we have extensively modified the material and methods section and have provided a full description of the methods that were missing (quantifications, circulating tumor cell detection etc.). Furthermore, we have eliminated irrelevant information, carefully described the control vectors and the breast cancer primary data sets, and revised and clarified the inconsistencies. Finally, we have re-written the statistical analysis section and revised all the references.

Point 2: *The authors present a very confusing introduction and an abstract that doesn't fully support the authors' claims e.g.*

- (a) *What is the evidence that "RARRES3 is negatively selected in tumors with high risk of lung metastasis?" what is the evidence that this is selection for RARRES3 rather than for subpopulations of cells?*
- (b) *What is the evidence that RARRES3 suppression facilitates reinitiation abilities at the metastatic site? The authors provide data that if the cells are inoculated directly into the lung that RARRES3 expression impairs the level of tumour burden in the lung but this is not an assay for reinitiation*
- (c) *How can RARRES3 be a target - you wouldn't want to target a metastasis suppressor*

Reply:

(a) We based our statement on the following facts:

- i) The significant association of RARRES3 low expression and risk of lung metastasis in breast cancer primary tumor samples (Figure 1 and Supplementary table 1). For each of these samples we had access to its gene expression profile based on Affymetrix arrays and the clinical annotation of time to metastasis.
- ii) Total RNA from these samples was isolated from 20-40 30- μ m thickness frozen sections of the tumor (100 mg). More than 50 per cent of the cells in each section were tumor cells. More details available at van de Vijer et al *NEJM* 2002, Wang et al *The Lancet* 2005 and Minn et al *Nature* 2005.

Nevertheless, we cannot exclude that the multiple sections pooled and analyzed per tumor may still represent a subpopulation of cells. Thus, following the reviewer's suggestion, we have modified the title and the abstract to avoid overstating our findings.

(b) In Figure 6A,B, we have provided evidence that cells (MDA-MB-231, CN37 and 4T1) expressing high levels of RARRES3 (exogenous or endogenous) are significantly less capable of initiating a lung metastatic lesion when implanted directly in the lung parenchyma. Moreover, pluripotent mammary epithelial cells and metastasis-initiating breast cancer cell populations can grow as floating aggregates, called mammospheres and oncospheres, respectively, when placed on non-adhesive plates in serum-free conditions — an experimental approach also used for subsequent confirmation of our observations (Figure 6C).

The standard approach to evaluating tumor-initiating capacity is based on testing the tumor-forming capacity of cellular populations at a range of dilutions after subcutaneous or orthotopic

injection into mice (i.e. Merlos-Suarez et al *Cell Stem Cell* 2011, Al Hajj et al *PNAS* 2003). However, at the distant site, cancer cells must remain viable as metastasis-initiating entities to eventually grow as overt metastatic lesions. Specialized microenvironments called metastatic niches define the outgrowth of metastatic nodules from metastasis-initiating cells. Importantly, when evaluating metastasis-initiating ability, other metastasis-related functions (homing, migration, adhesion, etc.) must be excluded. To this end, we injected the cells directly into the lung parenchyma (absence of extravasation/adhesion) as opposed to subcutaneously or in the mammary fat pad, since the extracellular matrix component of the lung metastatic niche has been reported to be important for lung metastasis-initiating capacity (Oskarsson et al *Nature Medicine* 2011). Grow emergence was used as a surrogate of the initiation capacity of cells, as per standard procedures (Merlos-Suarez et al *Cell Stem Cell* 2011, Al Hajj et al *PNAS* 2003).

We have modified the text to include all the new evidence collected and to clarify this issue (page 13 and onwards).

(c) We may not have explained ourselves properly. Our data suggest that activation of RARRES3 contributes to limiting metastasis initiation/progression. Thus, screening for compounds that activate RARRES3 expression or function may contribute to the development of new strategies to target tumor progression, as proposed in the discussion section. Following the reviewer's suggestion, we have removed this sentence from the abstract.

Point 3: *The authors show that RARRES3 can interact with MTDH (Figure 4E,F) but then never follow this up. Either they should demonstrate that this interaction is functionally important or remove this data.*

Reply: Following the suggestion of this reviewer and those of reviewers #2 and #3, we have focused on the novel RARRES3 metastasis suppressor function and lung metastasis initiation part of the original manuscript and strengthened the cell biology aspects of this section. The functional consequences of the interaction of RARRES3 with MTDH and changes in adhesion ability will be explored in an independent manuscript.

Point 4: *Why did the authors investigate PPAR target genes?? Again, not clear where this part of the manuscript is going*

Reply: The PLA_{1/2} catalytic activity of RARRES3 is central to its function. PLA_{1/2} enzymes are catalytically responsible for the production of arachidonic acid, which is subsequently processed to produce prostanoids and leukotrienes. These factors modulate several tumor functions and they are also key molecules in the regulation of differentiation and stem cell homeostasis. Well-known targets of these signal mediators are PPARs, a family of transcription factors previously reported to be crucial for cellular differentiation.

Together, the above observations set the stage to draw an association between RARRES3 and differentiation attributes. To this end, we confirmed that the levels of RARRES3 PLA_{1/2} were correlated with high levels arachidonic acid (Figure 5C) and with PPAR transcriptional activity (Figure 5D). Moreover, we show a significant correlation between PPAR target genes and

RARRES3 expression in primary tumors and low risk of lung metastasis (Figure 5E, F). On the basis of this set of observations, we further evaluate the association of RARRES3 expression and GATA3/GATA2 transcription factors in primary tumors (Figure 5G). Collectively, these findings prompted us to subsequently test the contribution of RARRES3 to metastasis initiation and cellular differentiation (Figure 6). We have modified the text accordingly to clarify this point.

Point 5: *A key conclusion from this manuscript is that RARRES3 expression "enforces breast cancer tumors to preserve their differentiation attributes". I may have been so muddled by this stage but it is not clear to me that the authors have been monitoring differentiation in their tumor cells nor the role of PPAR in this process.*

Reply: In the revised version of the manuscript, we have consistently determined the expression of pro-differentiation factors such as the transcription factor GATA3, as well as some well-known PPAR target genes. We have monitored the pro-differentiation consequences of RARRES3 expression in oncospheres and in lung metastasis lesions originated from cells expressing different levels of RARRES3 (Figure 6E,F). Moreover, we have measured these breast cancer cells capacity to initiate metastatic lesions (Figure 6A,B) and spheroid structures *in vitro* (Figure 6C,D). These key findings have been subsequently confirmed and monitored by testing the capacity of RARRES3 to differentiate RW4 mouse pluripotent embryonic cells and express differentiation markers (Figure 6G). We have changed the text to report all the new evidence collected.

Point 6: *Figure 2B. The authors state that they remove the tumors after 28 days, when the tumors had approximately 300 mm³. According to their growth curves in 2B the tumors had a size of approximately 100-150 mm³*

Reply: We apologize for the confusion. The tumor growth curves in 2D (formerly 2B) represent a different and independent experiment from that described in Figure 2A, B, C. This explains why tumor growth expands up to a volume of 600 mm³ in Figure 2D. On the contrary, to measure metastasis from the primary site, as described in Figure 2A tumors were resected when reaching 300 mm³. There was a delay of a few days between the two experiments, which is common between independent experiments separated in time (implying different animal batch, different serum batch, etc.). Of note, no differences in growth were observed between Mock and RARRES3-expressing cells in each independent experiment. We have modified the order of the figures and modified the text accordingly to clarify the independent experiments.

Point 7: *The authors show a very strong in vivo and ex vivo IVIS images for the LM2 lungs in Figure 2C. However, in the IHC vimentin staining in Figure 2D only single cells are found. According to the IVIS picture, you would expect proper mets and clusters in the lungs. The authors should show matched IVIS and IHC images from the same set of lungs. i.e. the images shown do not seem to be representative. In addition, higher power images of Figure 2D would be helpful*

Reply: Following the reviewer's suggestion, we have included new images to support our observation. In this experimental setting, we observed several small metastatic foci throughout

the lungs of the mice; however, given the short span of time from primary tumor resection, these lesions did not become large. We have included an image *inset* to provide further clarification of the type of lesions observed. Of note, the photon flux values represent absolute numbers of the amount of cells in all the lesions in the lung of the animal (integration of all 3 dimensions), which are projected in 2D images, as shown in Figure 2B, right panel. In contrast, Vimentin IHC images are a 2D representation of one section (3 μm) of the lungs, thus, while complementary, a direct comparison between the two systems is not possible.

Point 8: *As far as I can make out, the authors compare LM2 cells versus LM2-RARRES3 cells. The control that is needed is LM2 cells transfected with empty vector.*

Reply: In all our studies, LM2 cells were transfected with the same empty vector used to overexpress RARRES3. We have modified the text and figures to clarify this point.

Point 9: *The authors show that LM2 and LM2-RARRES3 cells injected into the MFP give rise to primary tumors with similar growth rates. Is the same true for the CN37a and CN37-shRARRES3 cells?*

Reply: We appreciate the reviewer's suggestion. We now provide the requested experiment. shControl, shRARRES3#1 and ShRARRES3#2 CN37 cells grew at similar rates when implanted in the mammary fat pad (Supplementary Figure S4D). We have modified the text accordingly to address this point in the revised manuscript.

Point 10: *Figure 4B. It is not possible to determine whether the cells have extravasated. The authors have flushed/perfused the lungs, but they should include vessel markers to exclude cells remained adhered to the inside of vessels.. What is the red stain - background???*

Reply: On page 20, materials and methods section, we have included a complete explanation of the procedure for lung extravasation experiments and the staining used.

In detail, CelltrackerTM green labelled cells were injected into mouse tail vein. Two days later, rhodamine-lectin was injected tail vein to label the vasculature, and 30 min later mice were perfused with 5 ml of PBS and sacrificed. Lungs were removed, trachea were perfused, and lungs were frozen in OCT. OCT sections were analyzed by confocal microscopy. The images show cells that have extravasated (green) and the vasculature (red). We have modified the text and figure legends to clearly address this point.

Point 11. *What do the red and green colors in Figure 5D represent. How were the oncospheres evaluated in Figure 6B?*

Reply: We appreciate the reviewer's suggestion. The red and green used in Figure 5E (formerly 5D) were redundant and have been eliminated.

We have included the description of the oncosphere evaluation in Figure 6D (formerly 6B) in the material and methods section, page 26, and the quantification procedure in the Figure 6D legend, page 39. In detail, in order to assess tumor initiation capacity *in vitro*, cells were counted and plated into low attachment 96-well plates at dilution of 1 cell per well and culture in Mammary Epithelial Basal Medium (MEBM, Lonza, Cat no. CC-3151), supplemented with MEGM SingleQuots (which contain Insulin, EGF, Hydrocortisone and GA-1000, LONZA cat no. 4136), 1X B27 without retinoic acid (GIBCO, Cat no. 12787-010), and 20 ng/mL of Recombinant Fibroblast Growth Factor (GIBCO, cat no. PHG0026). They were then incubated in 5%CO₂ at 37°C in order to obtain a first generation of oncospheres (anoikis and pluripotency selection) after 15 days. The process was repeated to ensure second-generation oncospheres (pluripotency selection). One plate per cell line was cultured. After two weeks of culturing, the oncospheres were counted under the microscope. The % of wells per plate (cell population) that generated oncospheres was quantified.

Reviewer #2

General Statement. *“Morales et al. clarify the role/molecular function of RARRES3, a previously identified lung metastasis suppressor gene, during pulmonary metastasis... Overall, the experiments performed are well executed and controlled and, in general, the authors’ conclusions are supported by their data.”*

Reply: We appreciate the reviewer’s statement.

Point 1: *Most of the studies relied on two cell lines, and it would be nice to use PDX’s and/or syngeneic GEM models with an intact immune system as well*

Reply: Following the suggestions made by this reviewer and reviewer #3, we have included new experiments using alternative cell models. These experiments corroborate our hypothesis. *In vivo* loss-of-function experiments using patient-derived breast cancer cells from pleural fluids (CN37 cells) confirmed that RARRES3 depletion supported lung colonization (Figure 3C and Supplementary Figures S4B,C,D). We also used these cells to determine the contribution of RARRES3 to lung metastasis-initiating property (Figure 6A,C and Supplementary Figure S7B). Furthermore, we have expanded our analysis using 4T1 murine breast cancer cells. These cells represent a spontaneous BALB/c murine breast cancer cell model, which makes them an ideal model for analysis in a syngeneic background with an intact immune system. To this end, 4T1 cells were transduced with a RARRES3 expression vector and their lung colonization capacity was examined 20 days post injection (Figure 3D and Supplementary Figure S4E). Subsequently, the metastasis-initiating capacity of these cell populations was tested in limiting dilution assays *in vivo* and in organotypic 3D cultures *in vitro* (Figure 6B,C and Supplementary Figure S7A,B)

We believe that the new experiments strengthen the observation that RARRES3 is functionally relevant for lung metastasis. We have modified the text and figures to include all the new observations.

Point 2: *The authors show that RARRES3 and MTDH can bind and that altering RARRES3 changes adhesion of cells to collagen I or fibronectin, but a strong connection of the two events is not made. For example, does MTDH overexpression rescue RARRES3-expression induced block in adhesion? How does the physical interaction of RARRES3 with MTDH disrupt its function during adhesion? Please expand on in the Discussion or provide experiments to elucidate the mechanism.*

Reply: The functional consequences of the RARRES3 interaction with MTDH and how this leads to the modification of MTDH adhesion properties is complex and has important implications. Following the suggestion of this reviewer and of reviewers #1 and #3, we focused on the novel RARRES3 metastasis suppressor function and lung metastasis initiation part of the original manuscript and strengthened the cell biological aspects. The functional implications of the RARRES3 interaction with MTDH and the changes in adhesion capacity will be developed in an independent manuscript.

Point 3: *Section title for second section of results currently reads "RARRES3 Promotes Breast Cancer Lung Metastasis". Perhaps this is a typo and should read "RARRES3 depletion promotes breast cancer lung metastasis" or "RARRES3 prevents breast cancer lung metastasis".*

Reply: We thank the reviewer for his/her observation and have modified the subheading accordingly.

Point 4: *While the authors do confirm that Aox-3x-PPRE-Luc reporter is responsive to RARRES3 catalytic activity, to claim that "PPAR transcriptional activity could be used as a surrogate marker of RARRES3 PLA1/2 activity" appears to be a bit of an overstatement. Many other cellular events may contribute to the final readout of the luciferase reporter.*

Reply: Following the reviewer's suggestion, we have modified the text to avoid overstating our findings.

Point 5: *Are arachidonic acid levels, prostinoids and/or leukotrienes reduced in the regions of downregulation of RARRES3 in human tumors or in vitro models? Does this correlate with outcome?*

Reply: We appreciate the reviewer's suggestion. We now provide the requested experiments in *in vitro* models. This includes arachidonic acid level measurements based on an ELISA assay corresponding to parental and LM2 MDA-MB-231 cells expressing different levels of RARRES3 (Figure 5C). The levels of arachidonic acid correlated with RARRES3 expression and inversely with cell capacity to colonize the lungs.

Referee #3

General Statement. *"The manuscript by Morales et al. investigates the contribution of RARRES3 in breast cancer metastases to the lung...*

These conclusions are generally carefully documented with specific consideration of alternative

mechanisms.”

Reply: We appreciate the reviewer’s statement.

Point 1. *The model is based on a single cell line, which is generally a weakness, but the data were substantiated by analysis of human tumors*

Reply: Following the suggestion made by reviewer #2, we have expanded our analysis to two independent cell models (including a syngeneic model), in addition to human tumor analysis to support the data. Please see the reply to point 1 of reviewer #2 for more details.

Point 2: *The paper is well-written overall but there are gaps in the data follow through that might be addressed, such as the importance of MTDH for phenotype. There are limiting weakness is the description of methods and lack of sufficient information about the size and replication of experiments, particularly cellular and in vitro data, on which conclusions are based. For example whether N= mice or fields or slides in Figure 3D. This information should be explicitly stated in figures, text or figure legends, including number of experiments for which a representative experiment is shown.*

Reply: Following reviewer’s suggestions, we have carefully revised the manuscript to address all the points detailed above and others that we have further encountered. In summary, we have modified the text to streamline the message, to provide format consistency, and to detail material and methods, figure legends, and the number of experiments for which a representative experiment is shown.

Moreover, based on the suggestion made by this reviewer and reviewers #1 and #2, we have focused on the novel metastasis suppressor function of RARRES3 and lung metastasis initiation part of the original manuscript and strengthened its cell biological aspects. The functional consequences of the RARRES3 interaction with MTDH and changes in adhesion capacity will be developed in an independent manuscript. We have modified the text of the revised manuscript accordingly and believe that this has largely improved the flow.

Point 3: *A confusing point is made on Pg 7, and again in Figure 3 C: the statement "CN37 patient-derived ER metastatic breast cancer cells' is a bit odd as it implies a primary or cells strain, i.e. recently patient-derived, but the sentence goes on to compare these cells to 'parental' well-established cancer cell line, MDA-MB-231. Please clarify.*

Reply: We have modified the text to clarify this point. In brief, CN37 cells were derived from metastatic pleural fluids of patients and used as an alternative cellular population to validate the contribution of RARRES3. To this end, we first compared the basal expression of RARRES3 between CN37 and MDA-MB-231 parental cells. We determined that both expressed similar levels of our gene of interest. Furthermore, RARRES3 loss-of-function experiments using CN37 cells were performed to confirm our hypothesis that RARRES3 was a causal metastasis suppressor.

Acceptance

23 April 2014

Thank you for the submission of your revised manuscript to EMBO Molecular Medicine. We have now received the enclosed reports from the referees that were asked to re-assess it. As you will see the reviewers are now globally supportive and I am pleased to inform you that we will be able to accept your manuscript pending the following final request.

Every published paper now includes a 'Synopsis' to further enhance discoverability. Synopses are displayed on the journal webpage and are freely accessible to all readers. They include a short standfirst - to be written by the editor - as well as 2-5 one sentence bullet points that summarise the paper (to be written by the author). Please provide the short list of bullet points that summarise the key NEW findings. The bullet points should be designed to be complementary to the abstract - i.e. not repeat the same text. We encourage inclusion of key acronyms and quantitative information. Please use the passive voice. Please attach these in a separate file or send them by email by replying to this message, we will incorporate them accordingly.

I look forward to receiving the bullet points as soon as possible.

***** Reviewer's comments *****

Referee #1 (Remarks):

In this revised submission, the authors present a much clearer and improved manuscript. Further, there is no question that the authors considered all of the original review comments and have addressed each one. have taken great consideration of all reviewers' comments and have addressed each one thoroughly.

The materials and methods are now complete and comprehensive, and all ambiguous statements have been carefully described and clarified. Furthermore there is a considerable amount of new experimental data including the addition of a different tumor model using 4T1 cells (figure 3 and 6). Pictures are now more representative of their quantification and better support authors' conclusions. The additional data further supports their findings and adds value to the manuscript.

The high quality and interesting results of this work, while already obvious in the previous submission, has now become much clearer and focused. The removal of the data with MTDH allows the results to be more streamlined without compromising the mechanistic findings.

I strongly support the publication of this manuscript and consider this revised version highly suitable for EMBO Mol Medicine.

Referee #2 (Remarks):

The authors have responded to the majority of the concerns of the reviewers and included additional data with two other cell line models to help provide additional support for their conclusions. The manuscript is much improved, although there still are a number of grammatical and spelling errors that need to be corrected.